



# Is it north or west foehn? A Lagrangian analysis of PIANO IOP 1

Manuel Saigger[1] and Alexander Gohm[1]

[1]Department of Atmospheric and Cryospheric Sciences (ACINN), University of Innsbruck, Innsbruck, Austria

**Correspondence:** Manuel Saigger (manuel.saigger@gmail.com)

**Abstract.** A case study of a foehn event in the Inn Valley near Innsbruck, Austria, is investigated that occurred on 29 October 2017 in the framework of the first Intensive Observation Period (IOP) of the Penetration and Interruption of Alpine Foehn (PIANO) field campaign. Accompanied with northwesterly crest-level flow, foehn broke through at the valley floor as strong westerly winds in the moring and was terminated in the afternoon by a cold front arriving from the north. The difference between local and large-scale wind direction rises the question of whether the event should be classified as north or west foehn – a question that has not been convincingly answered in the past for similar events based on Eulerian approaches. Hence, the goal of this study is to assess the air mass origin and the mechanisms of foehn penetration to the valley floor based on a Lagrangian perspective. For this purpose a mesoscale simulation with WRF and a backward trajectory analysis with LAGRANTO are conducted.

The trajectory analysis shows that the major part of the air mass arriving at Innsbruck originates six hours earlier over eastern France, crosses the two mountain ranges of the Vosges and the Black Forest and finally impinges on the Alps near Lake Constance and the Rhine Valley. Orographic precipitation over the mountains leads to a net diabatic heating of about 2.5 K and to a moisture loss of about 1 g kg$^{-1}$ along the trajectories. A secondary air stream originates further south over the Swiss Plateau and contributes with about 10 to 40 % to the foehn air in Innsbruck. Corresponding trajectories are initially nearly parallel to the northern Alpine rim and get lifted above crest level in the same region as the main trajectory branch. Air parcels within this branch experience a net diabatic heating of about 2 K, and, in contrast to the ones of the main branch, an overall moisture uptake due to evaporation of precipitation formed above this air mass. Penetration into the Inn Valley mainly occurs in the lee of three local mountain ranges – the Lechtal Alps, the Wetterstein, and the Mieming Chain – and is associated with a gravity wave and a persistent atmospheric rotor. A secondary penetration takes place in the western end of the Inn Valley via the Arlberg Pass and Silvretta Pass. Changes in the upstream flow conditions cause a shift in the contributions of the associated penetration branches. From a Lagrangian perspective this shift can be interpreted on the valley scale as a gradual transition from west to northwest foehn, despite the persistent local west wind at Innsbruck. However, a clear classification in one or the other category remains subjective even with the Lagrangian approach and, given the complexity of the trajectory pattern, is nearly impossible with the traditional Eulerian view. Likewise, foehn criteria based on pure adiabatic heating due to subsidence on the leeward side, i.e., the isentropic drawdown mechanism, are not appropriate to classify such moist events.



# 1 Introduction

The influence of mountains on the atmosphere comprises a large range of phenomena and scales. For example, a large-scale flow impinging on a mountain range may lead to upstream blocking and deflection of the low-level flow, modification of the precipitation distribution, excitation of gravity waves and the formation of downslope windstorms on the leeward side (Jackson et al., 2013). Downslope windstroms, such as the Alpine foehn, have large impacts on infrastructure and aviation savety due to their high wind speeds, strong turbulence and associated rapid air mass changes (e.g., ICAO, 2005; Gohm et al., 2008; Sharman and Lane, 2016). Apart from that, they also have been shown to influence air quality within foehn valleys (e.g., Seibert et al., 2000; Gohm et al., 2009; Li et al., 2015).

One prominent region of foehn research has been the Inn and Wipp Valley around Innsbruck, Austria, with a research history dating back to the 19th century (Seibert, 2012). Since then almost all the research in this area has been focusing on *south* foehn (e.g., Zängl, 2003; Mayr et al., 2007; Umek et al., 2021a), i.e., a flow originating south of the Alps, passing the main Alpine crest at the Brenner Pass and reaching Innsbruck in the west–east aligned Inn Valley via the south–north aligned Wipp Valley (Fig.1c). Although it is known since the early 20th century that foehn can reach Innsbruck also from the north (Hann, 1891; Trabert, 1903), there has been only very little research on the topic since then. Apart from these old publications, published research comprises an observational case study by Wankmüller (1995), a climatology by Haas (2006), and one case study by Zängl (2006) based on numerical simulations. This limited amount of research is the reason that there is still a lack in the understanding of the phenomenon. Especially the origin of the air mass and the penetration mechanisms of the foehn into the Inn Valley are still widely unclear (Zängl, 2006). This knowledge gap also impedes a clear-cut classification into *north* foehn in case of large-scale *northerly* flow above crest level but *westerly* flow in the Inn Valley.

A peculiarity about this type of foehn in Innsbruck is that it barely ever reaches the valley bottom directly from north via the mountain range of Nordkette (Haas, 2006, see Fig. 1c for the location). The height and steepness of this mountain range, the lack of topographic gaps, and the stable stratification of the valley atmosphere usually prevents penetration of the flow directly form the north to the valley floor. More often the flow gets channeled by the Inn Valley and appears as a strong westerly to northwesterly wind in Innsbruck (Haas, 2006). As a result of this channeling by the Inn Valley, different authors have used different expressions and different definitions for the phenomenon. Trabert (1903) and Wankmüller (1995) called the phenomenon *northwest* foehn and *north* foehn, respectively, mainly based on the measured *local* wind direction in Innsbruck. Investigating the same case as Wankmüller (1995), Zängl (2006) used the expression *north* foehn, but suggested that the case should rather be referred to as *northwest* foehn, due to the absence of a cross-Alpine gradient in pressure or temperature. Haas (2006) used a definition pointing into the same direction, making a distinction between *north* and *west* foehn based on whether or not there is a cross-Alpine flow. This was decided based on the wind direction and speed in the Wipp Valley both at the valley bottom and at crest level, as well as on the pressure difference between the northern Alpine foreland and Innsbruck. Apart from these local and large-scale approaches, the more intuitive approach based on air mass origin and prominent locations of foehn penetration has not been considered so far. For the sake of simplicity we will refer to this phenomenon as north foehn in the





remainder of this section. However, we will show later whether this description is appropriate for the foehn event examined in

this study.

According to Haas (2006), the relative frequency of north foehn in Innsbruck is about 1 % derived from 10 min observations recorded over a period of 5 years (1.3 % at the station Innsbruck Airport, 0.7 % at the station Innsbruck University). Therefore, north foehn occurs less frequently than south foehn, which has a relative frequency of about 5 % (Föst, 2006). North foehn in Innsbruck exhibits a pronounced annual and daily cycle with three frequency peaks in the afternoon hours of late winter, late

autumn, and early summer (Haas, 2006). This frequency pattern can also be found for north foehn at stations south of the main Alpine crest (Verant, 2006; Cetti et al., 2015).

Using numerical simulations, Zängl (2006) confirmed the aforementioned channeling of the foehn flow in the Inn Valley. Based on an Eulerian model perspective, he speculated that the foehn entered the Inn Valley via the Arlberg Pass and the Silvretta Pass at the western end of the Inn Valley, and possibly also in the lee of the Lechtal Alps. As already found for south

foehn, vertically propagating large-amplitude gravity waves appeared to play an essential role in the penetration of the flow into the valley.

Despite the striking features in Zängl's simulations, a pure Eulerian view on the wind field does not allow for definite statements on the air mass origin. This gap can be filled, e.g., by a mass flux analysis (e.g., Dautz, 2010; Arduini et al., 2020; Sabatier et al., 2020), or by a trajectory analysis. Especially trajectories have revealed complex three-dimensional flow patterns

in the context of foehn winds with different flow branches upstream of the mountains (e.g., Elvidge et al., 2015; Takane et al., 2015; Würsch and Sprenger, 2015; Miltenberger et al., 2016; Jansing and Sprenger, 2020). By analyzing the heat and moisture budgets along the trajectories, the importance of various diabatic and adiabatic processes within the flow can be assessed. Based on this methods, several studies have detected source regions of moisture (Langhamer et al., 2018; Schuster et al., 2021) and quantified the contribution of different processes to heat events (Ishizaki and Takayabu, 2009; Takane et al., 2013, 2015)

and melt events (Elvidge et al., 2015; Zou et al., 2019; Hermann et al., 2020; Zou et al., 2021). By separating the individual terms in the temperature tendency equation, Miltenberger et al. (2016) were able to assess the role of diabatic and adiabatic processes to foehn warming.

Apart from the uncertainties in the origin of the air mass, there is some disagreement on the role of precipitation in the development of north foehn. Haas (2006) found that in 20 % of the times north foehn in Innsbruck was associated with

measurable precipitation. In 40 % of the times precipitation occurred in Kufstein (KUF, see Fig. 1b), located approximately 80 km east of Innsbruck at the northern edge of the Alps. In contrast to that, Zängl (2006) showed with simulations using full and modified model physics that orographic precipitation on the northern side of the Alps inhibited the formation of north foehn in the Inn Valley. Zängl (2006) reasoned that precipitation falling into layers of unsaturated air in the valley stabilized these layers due to evaporative cooling. This dampened the amplitude of gravity waves and thus prevented the foehn from

penetrating into the valley.

Also in a broader perspective moist processes have been shown to add a lot of complexity to foehn-like flows and mountain waves. Due to the release of latent heat in a saturated environment, the effective moist Brunt-Väisälä frequency $N_m$ is lower compared to the dry case (Durran and Klemp, 1982b). Depending on the stability, the location and the amount of moisture in



certain layers, Durran and Klemp (1982a) found numerous different effects on mountain waves. These included an increase in
horizontal wave length and damping or amplification of various wave modes, depending on the exact profiles of moisture, temperature and wind. In some cases even instability was released. Gravity wave amplitude decreased in the simulations of Durran and Klemp (1983) with the presence of moisture, leading to a weakening of the associated downslope windstorm. Similar to Zängl (2006), Zängl and Hornsteiner (2007) found that orographic precipitation seemed to suppress foehn development.

Latent heat release on the windward side due to orographic lifting decreases the stability of the air mass. This may even
prevent low-level flow splitting/blocking on the windward side by prompting flow over instead of flow around the mountain (e.g., Buzzi et al., 1998; Schneidereit and Schär, 2000; Rotunno and Ferretti, 2001; Jiang, 2003). Smith et al. (2003) and Miltenberger et al. (2016) identified a phenomenon which they termed "air mass scrambling". In such a situation air parcels that gain latent heat on the windward side while rising over the mountain are too buoyant to descend substantially on the leeward side and remain at high altitudes. In contrast, parcels starting at higher levels that experience diabatic cooling strongly
descend on the leeward side and end up at low levels.

The field campaign of the research project Penetration and Interruption of Alpine Foehn (PIANO) focused on the investigation of south foehn (e.g., Haid et al., 2020; Muschinski et al., 2020; Haid et al., 2021; Umek et al., 2021a). However, the first event observed on 29 October 2017 during the first Intensive Observation Period (IOP 1) was no south foehn. Accompanied by a very strong northwesterly synoptic-scale flow, this special type of foehn established as strong westerly winds in the Inn
Valley and prevailed at Innsbruck between 0800 and 1530 UTC. The rich observational data set of PIANO, in combination with mesoscale numerical simulations and trajectory analysis, provides a great opportunity to investigate this special type of foehn in detail. The main research questions are:

–  Where does the foehn air reaching Innsbruck come from on a regional scale before impinging on the Alps?

–  Where and how does the foehn penetrate into the Inn Valley?

–  What is the role of adiabatic and diabatic processes both on the regional and the local scale?

–  Does the trajectory analysis help to classify the case as north or west foehn?

Section 2 will describe the observational data set used in this study, as well as the setup of the numerical model, and the method of the trajectory analysis. After an overview of the meteorological evolution on the large and local scale in Section 3, the regional aspects of the air mass origin will be investigated in Section 4, while Section 5 focuses on the local aspects of the
penetration mechanisms. The results will be interpreted and discussed in the context of earlier investigations in Section 6.

## 2   Data and methods

### 2.1   PIANO campaign

The field campaign of the PIANO project took place in autumn and early winter of 2017 in the Wipp Valley and Inn Valley. Foehn was observed during seven Intensive Observation Periods (IOPs). The first IOP is the subject of this study. Detailed

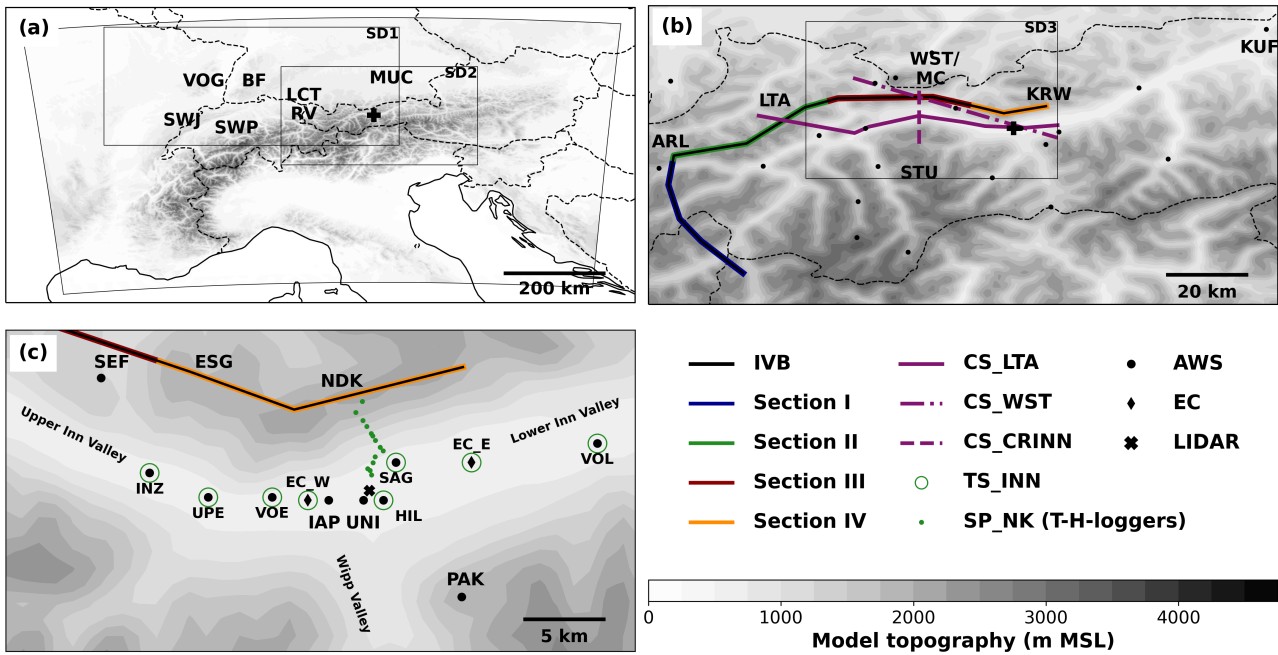

**Figure 1.** WRF model topography of the target area (contours, 250 m increments): (a) The full model domain, (b) the Inn Valley, and (c) the region of Innsbruck. Abbreviations describe prominent locations: Swiss Jura (SWJ), Swiss Plateau (SWP), Vosges (VOG), Black Forest (BF), Lake Constance (LCT), Rhine Valley (RV), Arlberg Pass (ARL), Lechtal Alps (LTA), Wetterstein and Mieming Chain (WST/ MC), Karwendel (KRW), Stubai Alps (STU), Kufstein (KUF), Seefeld (SEF), Erlspitze Group (ESG), Nordkette (NDK), Patscherkofel (PAK), Innsbruck Airport (IAP), University of Innsbruck (UNI). Innsbruck is marked by a black cross in (a) and (b). The black line (IVB) in (b) and (c) marks the northern boundary of the Inn Valley represented by the crest line between ARL and KRW as well as the western boundary of the Inn Valley south of ARL. Colored lines indicate the subsections of IVB. Three different purple lines in (b) illustrate the locations of vertical cross sections. Locations of observation sites are indicated by black symbols in (b) and (c) according to the observation type; abbreviations indicate station names. Transects and slope profiles are indicated by green symbols: open circles for valley transect TS_INN, dots for the slope profile along NDK (SP_NK). National borders are depicted as dashed black lines in (a) and (b). Subdomains SD1, SD2, and SD3 are indicated as black rectangles in (a) and (b).

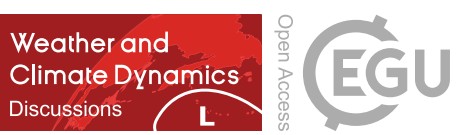

information about the field experiment and its instrumentation can be found, e.g., in Muschinski et al. (2020), Haid et al. (2020), and Haid et al. (2021). The observations used in the closer surrounding of Innsbruck that were used in this study are depicted in Fig. 1c. Of special interest for model verification were the automatic weather stations (AWS) at the Universtiy of Innsbruck (UNI, located at 578 m above mean sea level, MSL) and at Innsbruck Airport (IAP, 578 m MSL). Additionally, surface stations along a valley transect were used (TS_INN in Fig. 1c). They consisted of the AWS at Inzing (INZ, 597 m MSL),

Unterperfuss (UPF, 594 m MSL), Völs (VOE, 583 m MSL), on the rooftop of the former hotel Hilton (HIL, 629 m MSL), in Saggen (SAG, 569 m MSL), and in Volders (VOL, 552 m MSL), as well as the eddy-covariance stations in the western (EC_W, 579 m MSL) and the eastern part of the city (EC_E, 562 m MSL). One slope profile consisting of temperature and humidity loggers covering the whole Nordkette (SP_NK) was used to characterize the stratification of the valley atmosphere. The corresponding temperature measurements were in good agreement with radiosoundings taken in the middle of the valley

(Haid et al., 2020). From the four Doppler wind lidars operated during the campaign, only data from one lidar (SL88, located on a rooftop in the city center) was used to derive vertical profiles of the mean horizontal wind following Haid et al. (2020). For a larger-scale picture, data from operational AWS distributed over Tirol and the German Alpine foreland as shown in Fig. 1b was analyzed.

Potential temperature was calculated in two different ways depending on the availability of pressure observations at a specific

site. For stations near the valley floor, which recorded temperature and pressure, the standard formula for potential temperature was used (e.g., Wallace and Hobbs, 2006):

$$\theta = T \left( \frac{p_0}{p} \right)^{\frac{R}{c_p}}. \tag{1}$$

Here $T$ is the measured air temperature, $p$ is the measured air pressure, and $p_0 = 1000$ hPa is the reference pressure. $R$ and $c_p$ denote the gas constant for dry air and the specific heat capacity of air at constant pressure, respectively. For stations along

the slope which did not record pressure, potential temperature was calculated with the same approach as in Muschinski et al. (2020) based on a reduction from station height $z$ to reference height $z_0$ rather than a reference pressure $p_0$:

$$\theta = T + (z - z_0)\Gamma_d. \tag{2}$$

Here $\Gamma_d$ is the dry adiabatic lapse rate with $\Gamma_d = 0.0098$ K m$^{-1}$. In order to minimize the difference between Eq. (1) and (2) we used a reference level of $z_0 = 0$ m MSL instead of the height of the valley floor used in Haid et al. (2020) and Muschinski

et al. (2020). Nevertheless, whenever comparing observations directly with the model output (see Sect. 3.2) we use the same formula for both observations and model data.

## 2.2 Setup of the numerical model

The numerical simulation in this work are carried out with the Advanced Research version of the Weather Research and Forecasting (WRF-AWR) model, version 4.1 (Skamarock et al., 2019). The setup of the model is close to the mesoscale

simulations of Umek et al. (2021a). The model consists of one single domain covering the entire Alpine region with a horizontal mesh size of $\Delta x = 1$ km and $1100 \times 750$ points. A hybrid sigma-pressure coordinate is used with 80 vertical levels. The lowest





model level is at about 10 m above ground level (AGL) and the model top is at 40 hPa which corresponds to a height of about 21 km. The vertical spacing increases from $\Delta z = 20$ m at the surface to $\Delta z = 400$ m at the model top, which results in about 33 vertical levels below crest height of the Inn Valley. The upper 8 km act as a damping layer (Klemp et al., 2008).

Model orography, soil data, and land use data are similar to Umek et al. (2021a) and are based on the Shuttle Radar Topography Mission (SRTM) digital elevation model (USGS, 2000), the Harmonized World Soil Database (HWSD; Milovac et al., 2014), and the CORINE Land Cover Inventory 2012 (European Environment Agency, 2017), respectively.

Turbulence in the planetary boundary layer (PBL) is parametrised with the Mellor-Yamada-Nakanishi-Nino (MYNN) level 3 scheme (Nakanishi and Niino, 2006) with the advection of TKE being activated. While vertical mixing is performed by the
PBL scheme, horizontal diffusion is calculated in physical space (*diff_opt* = 2) based on a first order closure after Smagorinsky (1963). The MYNN surface-layer scheme (Nakanishi and Niino, 2006) and the unified Noah land-surface model (Ek et al., 2003) is employed. For short and longwave radiation the RRTMG scheme after Iacono et al. (2008) is used and topographic shading as well as slope effects on radiation are activated. Cloud microphysics are parametrised with the Thompson scheme (Thompson et al., 2008). Convection is assumed to be fully resolved in the model and, thus, is not parametrised.

The operational 6-hourly high-resolution analysis of the Integrated Forecasting System (IFS) of the European Centre for Medium Range Weather Forecasts (ECWMF) is used as initial and boundary conditions. In order to correct the erroneous snow cover distribution in the ECMWF analysis, all snow below 1800 m MSL was removed. This height of the snow line was determined from webcam images of the region west of Innsbruck. Extrapolation of the temperature field of the ECMWF analysis to WRF model levels located below the IFS topography is done with a moist adiabatic lapse rate of $\Gamma = 6.5$ K km$^{-1}$.
In contrast to Umek et al. (2021a) no observation nudging was employed.

The simulation starts at 12 UTC 28 October 2017 and ends at 00 UTC 30 October 2017 covering the whole timespan of the foehn event with a spin-up time of several hours before the main foehn period starts. In addition to the standard model output interval of one hour, a finer output interval of 5 minutes is used for the main foehn phase between 00 UTC and 15 UTC 29 October 2017. These fields serve mainly as input for the trajectory calculations and include the three-dimensional fields of all
three wind components, temperature, pressure, geopotential height, mixing ratios of water vapor and all hydrometeor classes, subgrid-scale turbulence kinetic energy (SGS-TKE), as well as accumulated precipitation at the surface.

### 2.3   Trajectory analysis

#### 2.3.1   LAGRANTO settings

The trajectory analyses are carried out with the Lagrangian Analysis Tool LAGRANTO version 2.0 (Sprenger and Wernli,
2015). Parcels for backward trajectories are released every hour between 06 and 15 UTC 29 October 2017 in a volume with a square base of 4×4 km centered over Innsbruck and a height of 1800 m covering the altitudes between 700 and 2500 m MSL. Starting points are equally distributed in this volume with a horizontal distance of 200 m and a vertical distance of 25 m, which results in a total number of 18688 trajectories per starting time. Trajectories are calculated backward in time for six hours. During these six hours the parcels have traveled far enough to determine the source regions upstream of the Alps but



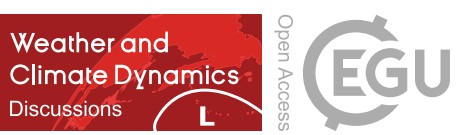

not too far to have left the model domain. The trajectory calculation is based on the WRF wind fields available every five minutes. The integration time step is 15 s and the output interval is 1 min. The jump option for trajectories intersecting with the model topography is active. Although the calculation of trajectories is numerically stable for all integration time steps, Stohl (1998) suggests that the CFL criterion (Courant et al., 1928) should be fulfilled. This ensures that no grid cell is skipped by the trajectory and thus all features resolved by the input wind data are represented in the trajectory. Therefore, with the chosen

integration time step of 15 s the CFL criterion is not violated for a horizontal grid spacing of 1 km and wind speeds smaller than 66 m s$^{-1}$. The 5 min input interval lies at the upper end of the range suggested, e.g., by Bowman et al. (2013) and Schär et al. (2020) for kilometer-scale simulations. Experiments with different input intervals and integration time steps showed the expected error growth with decreasing temporal resolution but did not reveal significant impact of the LAGRANTO settings on the general conclusions of this study (Saigger, 2021).

In contrast to other trajectory analysis tools, LAGRANTO does not account for the impact of subgrid-scale turbulence on the trajectories. The calculation is solely based on the resolved wind field. Hence, the trajectories in the channeled flow in the Inn Valley below crest height are most likely less dispersive than in reality. Sensitivity experiments showed that a large number of trajectories distributed over a whole volume is necessary to capture at least the resolved trajectory dispersion (Saigger, 2021). Along the trajectories temperature, pressure, wind components, water vapor mixing ratio, hydrometeor mixing ratios, and TKE

are traced. Additionally also the precipitation intensity at the ground below the trajectory and the terrain height are saved.

### 2.3.2   Additional calculations

Additionally to the variables directly traced along the trajectories several other variables are diagnosed from the primary ones. The difference between the fields of accumulated precipitation of two following timesteps is calculated to obtain the 5 min precipitation intensity which is then multiplied by a factor of 12 to get a hourly precipitation intensity for easier interpretation.

A total water mixing ratio $q_t$ is derived by adding up the mixing ratios of water vapor and all six hydrometeor classes of the Thompson scheme. Ice-liquid potential temperature is calculated following Curry (2015),

$$\theta_{il} = \theta \exp\left(-\frac{L_v q_l}{c_p T} - \frac{L_s q_s}{c_p T}\right), \tag{3}$$

where $L_v$ and $L_s$ are the latent heat of evaporation and sublimation, $q_l$ and $q_s$ are the mixing ratios of liquid and solid water, respectively. The ice-liquid potential temperature is defined as the potential temperature that an air parcel would have if all

liquid water was evaporated and all ice particles were sublimated. For moist reversible processes, e.g., cloud formation and cloud dissipation without precipitation, $\theta_{il}$ stays constant. Hence, a decrease in total water content increases $\theta_{il}$ and vice versa. It is noteworthy that Eq. (3) is an approximate form and is only valid at the triple point (Curry, 2015). More accurate formulations of $\theta_{il}$, e.g., for applications in deep convection, are available (e.g., Bryan and Fritsch, 2004), which are computationally more expensive and, hence, are not used in this study. However, Curry (2015) states that $\theta_{il}$ based on Eq. (3) is an economical and

not too inaccurate way to treat ice processes in a numerical cloud model.





Analog to Hermann et al. (2020) the adiabatic and diabatic effects on temperature along the trajectory are calculated. By taking the logarithm and the material derivative of Eq. (1) one obtains an equation for the Lagrangian temperature tendency

$$\frac{DT}{Dt} = \frac{\kappa T \omega}{p} + H \left( \frac{p_0}{p} \right)^{-\kappa} . \tag{4}$$

Here the first term of the right-hand side describes the adiabatic heating or cooling along the trajectory with the Lagrangian
pressure tendency $\omega = \frac{Dp}{Dt}$ and $\kappa = R/c_p = 0.286$. The second term represents the diabatic heating or cooling with $H = \frac{D\theta}{Dt}$. Following Hermann et al. (2020) Eq. (4) can be discretized as

$$\left. \frac{DT}{Dt} \right|_{t=t_i} \approx \frac{T_i - T_{i-1}}{\Delta t} \approx \kappa \left( \frac{T_i + T_{i-1}}{p_i + p_{i-1}} \right) \left( \frac{p_i - p_{i-1}}{\Delta t} \right) + \frac{\theta_i - \theta_{i-1}}{\Delta t} \left( \frac{2p_0}{p_i + p_{i-1}} \right)^{-\kappa} , \tag{5}$$

where variables with subscript $i$ refer to time $t = t_i$ and subscript $i-1$ to time $t = t_i - \Delta t$, respectively. $\Delta t$ denotes the time interval between the individual trajectory points, in our case $\Delta t = 1$ min which represents the output interval (see Sect. 2.3.1).
In contrast to Hermann et al. (2020) the adiabatic term (first term on right hand side of Eq. (5)) is calculated explicitly and not as a residual term.

The spatial distribution of trajectory properties is determined by interpolating these properties from the trajectory position to a Cartesian grid with a horizontal mesh size of 3 km. This mesh size is larger than the grid point distance of the WRF domain in order to ensure that trajectories do not "skip" grid boxes between two time steps at which the trajectory positions are
known (1 min interval). For the calculation of the trajectory density the proportion of trajectories with at least one point within the grid box is determined. If a certain trajectory is detected more then once in the same grid box, e.g., at subsequent times steps, it is only counted once. Similarly mean trajectory properties are calculated on the same grid by taking the median of various parameters (e.g., trajectory height) over all trajectories within one grid box. To ensure that all trajectories are weighted equally independent on the number of points within the grid box, first the median for all points of one trajectory is calculated.
Subsequently, these median properties are averages over all trajectories. For temperature and moisture variables not only the local value but also the difference to the value at the arrival in Innsbruck is calculated and gridded in order to quantify heat and moisture gain or loss along the trajectories.

An appropriate valley boundary has to be defined in order to determine the point where the trajectories penetrate into the Inn Valley (see line IVB in Fig. 1b, c). In the north of the Inn Valley this boundary essentially follows the crest line. South of the
Arlberg Pass the line IVB is defined by the western end of the Inn Valley and its tributaries. For each trajectory the point where the valley boundary is first crossed is determined with a resolution of 1 km along the valley boundary. For later investigations the valley boundary is divided into four sections with distinct topographic features (see Fig. 1b, c). Section I covers the western end of the Upper Inn Valley, the Paznaun Valley with the Silvretta Pass, and the Arlberg Pass. Section II contains the ridge line of the Lechtal Alps until the region of Fern Pass. Section III covers the two mountain saddles of Fern Pass and Seefeld with
the Wetterstein Mountains and Mieming Chain in between, while Section IV stretches out over the Karwendel and includes its most western part of the Erlspitze Group (ESG) next to Seefeld as well as the Nordkette directly north of Innsbruck (NDK, see Fig. 1c).

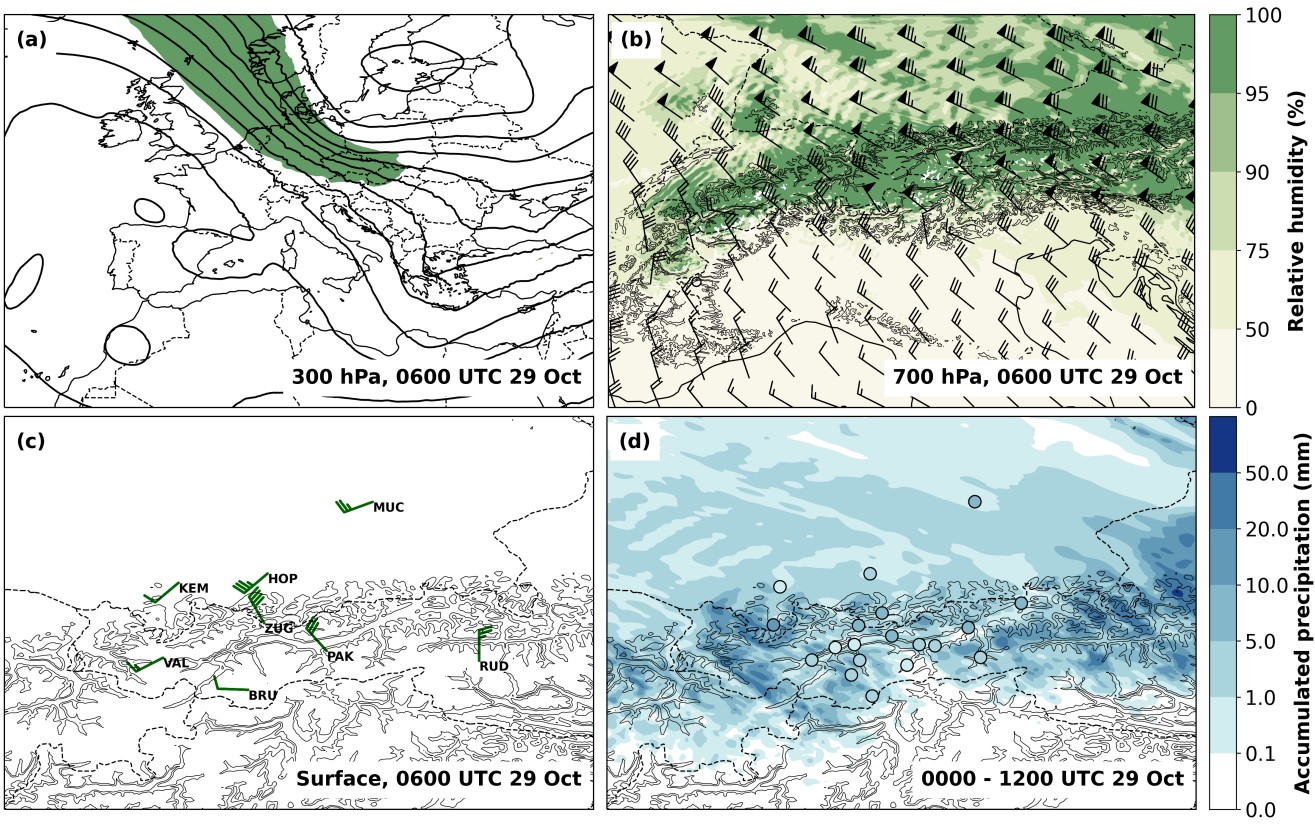

**Figure 2.** Synoptic situation on 29 October 2017. (a) ECMWF analysis of geopotential (black contour lines, spacing 1000 m$^2$ s$^{-2}$) and wind speed (shading, wind speed larger 50 m s$^{-1}$ is shaded in green) at 300 hPa at 0600 UTC. (b) WRF simulated fields of wind (barbs) and relative humidity (color contours) at 700 hPa at 0600 UTC. (c) Observed winds (barbs) at 0600 UTC. Abbreviations indicate station names: Munich (MUC), Kempten (KEM), Hohenpeissenberg (HOP), Zugspitze (ZUG), Valluga (VAL), Brunnenkogel (BRU), Patscherkofel (PAK), Rudolfshütte (RUD). (d) Precipitation accumulated between 0000 and 1200 UTC in subdomain SD2 simulated by WRF (color contours), and observed (filled circles with the same color scale). National borders as in Fig. 1, model topography in (b) - (d) is indicated by black contour lines for the elevations of 1000 and 1500 m MSL. Wind barbs in (b) and (c) indicate the wind speed in kn: half barb, full barb and triangle denote 5, 10 and 50 kn.
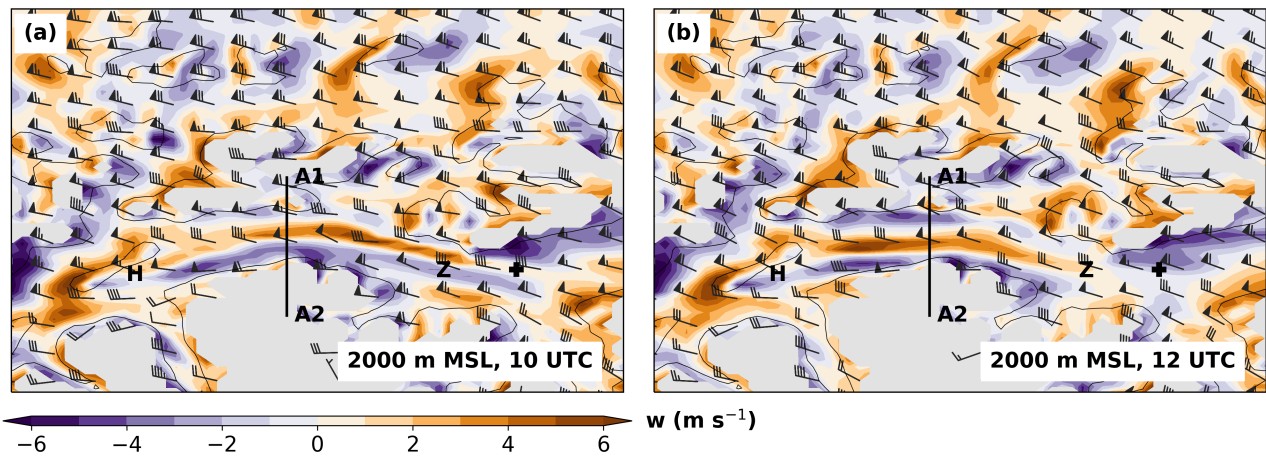

**Figure 3.** Simulated horizontal wind and vertical velocity at 2000 m MSL at (a) 1000 UTC and (b) 1200 UTC 29 October 2017 in subdomain SD3. Horizontal wind is depicted by wind barbs as in Fig. 2b, vertical velocities are illustrated by color contours. Model topography is indicated by black contour lines for the elevation of 1500 m MSL, areas with model topography higher than 2000 m MSL are shaded in grey. Locations of Haiming (H) and Zirl (Z) are indicated by letters, the location of Innsbruck is indicated by a black cross. The orientation of the vertical cross section CS_CRINN is depicted by a thick black line with A1 and A2 being the start and end points of the cross section as in Fig. 4.

## 3 Meteorological overview

### 3.1 Synoptic and mesoscale analysis

On 29 October 2017 a pronounced low-pressure system was located over northeastern Europe (Fig. 2a). The associated jet stream led to strong northwesterly flow towards the Alps. The lower troposphere north of the Alps was characterized by a pronounced elevated temperature inversion at around 1500 m MSL as depicted by the radiosonde profile of Munich at 00 UTC (not shown). Below this inversion the flow was deflected eastward by the Alps, while above crest level the flow crossed the Alps (Fig. 2c). The deflected flow is depicted in Fig. 2c by the observations at Munich and Kempten as well as at the station of

Hohenpeissenberg, a 977 m MSL high mountain in the Alpine foreland. The mountain top stations of Zugspitze, Patscherkofel and Rudlofshütte illustrate strong cross-Alpine flow with strong north to northwesterly winds, while for stations like Valluga and Brunnenkogel local flow deflection may explain the stronger westerly wind component. With the pressure low moving south and an embedded short-wave trough approaching the Alps, crest-level winds further intensified during the morning of 29 October 2017 with mean wind speeds of about 20 m s$^{-1}$ at Zugspitze and around 15 m s$^{-1}$ at Patscherkofel. Orographic

precipitation on the northern side of the Alps already occurred in the night and the morning before the cold front arrived in the afternoon (Fig. 2d). The high precipitation intensities were restricted to the stations at the very northern edge of the Alps





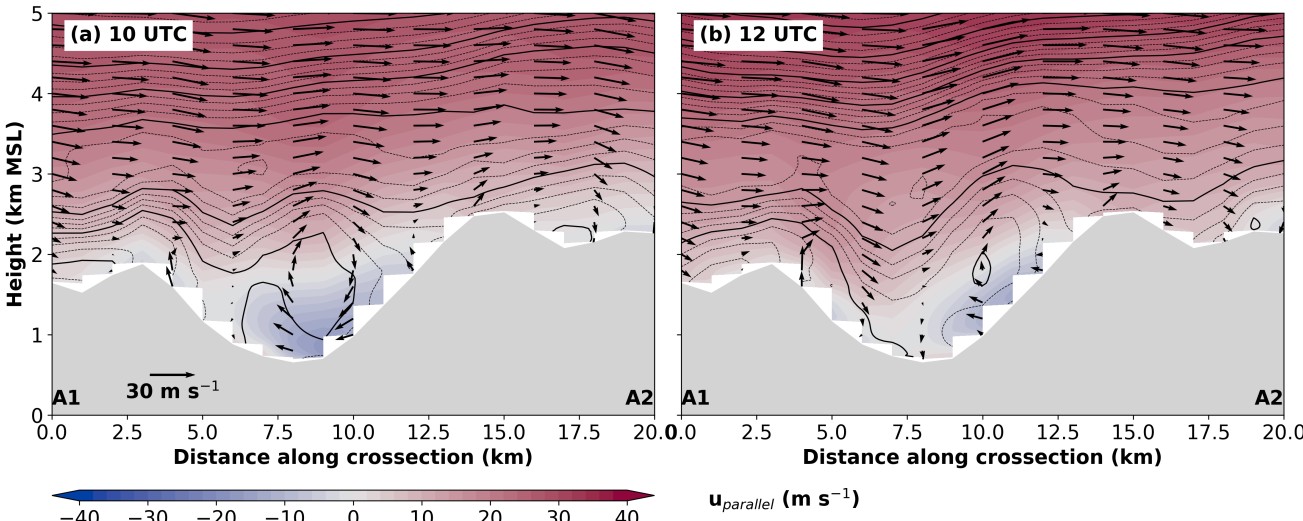

**Figure 4.** Vertical transect of simulated equivalent potential temperature (black contour lines, spacing 2 K for thick solid lines, spacing 0.5 K for thin dashed lines) and wind field (color contours and arrows) along cross section CS_CRINN at (a) 1000 UTC and (b) 1200 UTC 29 October 2017. Color contours represent the horizontal wind component parallel to the plane (positive towards right), arrows illustrate the two-dimensional wind field on the vertical plane. Model topography is shaded in grey. A1 and A2 in (a) and (b) mark the start and end point of the transect as depicted in Fig. 3.

with locally more than 10 mm of precipitation between 00 and 12 UTC 29 October 2017. A clear precipitation shadow can be detected in the Inn Valley with observed precipitation not exceeding 5 mm during the same period.

The WRF simulation reproduces the deflected flow at lower levels and the near-crest level flow crossing the Alps reasonably
well (Fig. 2b). As in the observations, strongest precipitation is simulated at the northern edge of the Alps as well as over the higher inner-Alpine mountain ranges (Fig. 2d). The simulated amounts are in the same range as the maximum observed values. The simulation also captures the rain shadow in the valleys.

The cross-Alpine flow formed a complex pattern of gravity waves. While during the first half of the night trapped lee waves with rather short wavelengths dominated in the Inn Valley region (not shown), horizontal wavelength increased in the second
half of the night. In the morning the crest-level flow field east of Innsbruck was dominated by a single gravity wave over the Inn Valley with subsiding motion on the northern slopes and rising motion on the southern slopes (Fig. 3a). In contrast, the flow field west of Innsbruck was dominated by an atmospheric rotor that was situated underneath a gravity wave and covered the whole valley cross section (Fig. 4a). This rotor led to subsiding motion on the southern side of the valley and rising motion on the northern side and stretched out in a roughly 30 km long section along the Inn Valley between Haiming and Zirl (see
dipole structure between H and Z in Fig. 3a). The rotor was dominant in this region between 06 and 11 UTC. Later, crest-level winds shifted to a slightly stronger northerly component (Fig. 3b) and the horizontal wave length and the amplitude of the lee



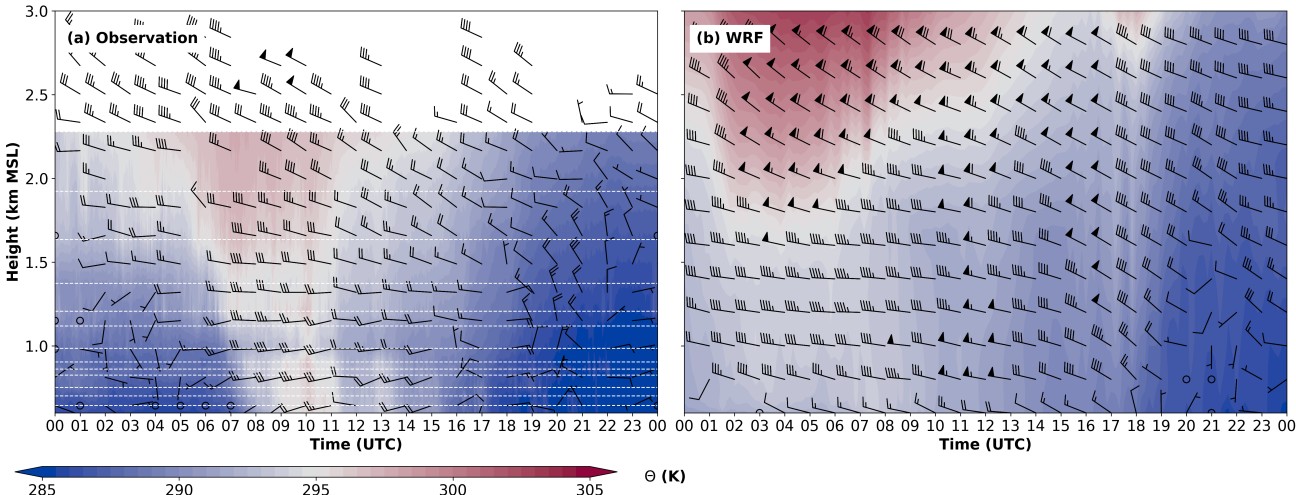

**Figure 5.** Time-height diagram of (a) observed and (b) simulated potential temperature based on Eq. (2) and mean horizontal wind at Innsbruck on 29 October 2017. Potential temperature is shown by color contours, wind barbs indicate wind speed and direction as in Fig. 2. Temperature measurements were conduced along the slope profile SP_NK, white horizontal lines indicate the measurement heights. Wind barbs are shown at an hourly interval and represent 10 min averages of the horizontal wind measured with the Doppler wind lidar SL88. Potential temperature and wind in (b) are 10 min averages of the time series of every integration time step at the WRF grid point closest to UNI.

wave increased. This led to a weakening and shift in the position of the rotor and to stronger subsidence on the northern slopes of the Inn Valley (Fig. 3b, Fig. 4b).

## 3.2 Local analysis

The ambient northwesterly flow was channeled into a westerly flow inside the valley (Fig. 5a). During the night the lower valley atmosphere was still decoupled from the flow aloft with low wind speeds and a stable stratification up to about 1500 m MSL. Light precipitation was measured at the surface (not shown). During the morning hours the foehn gradually descended down and broke through in Innsbruck at the valley floor at about 08 UTC (Fig. 5a). The valley atmosphere was still slightly stably stratified based on dry potential temperature. The foehn breakthrough occurred two hours earlier in the west of Innsbruck at

Inzing and subsequently progressed eastward (Fig. 6a). After around 10 UTC the whole valley atmosphere was well mixed (Fig. 5a). The vertical profile of the horizontal wind above Innsbruck exhibited a low-level jet stucture with a maximum at around 800 m MSL and a local minimum at about 1200 m MSL. (Fig. 5a). After around 11 UTC winds above crest height weakened slightly and with the advection of potentially colder air also the westerlies inside the valley lost strength. At the surface this cooling manifested itself with two distinct drops in temperature at about 1130 and 1330 UTC before the arrival of

the cold front at 1530 UTC (Fig. 7a). The cold front arrived in Innsbruck from the lower Inn Valley (Fig. 6a), filling the valley


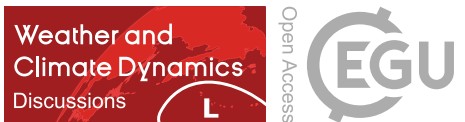

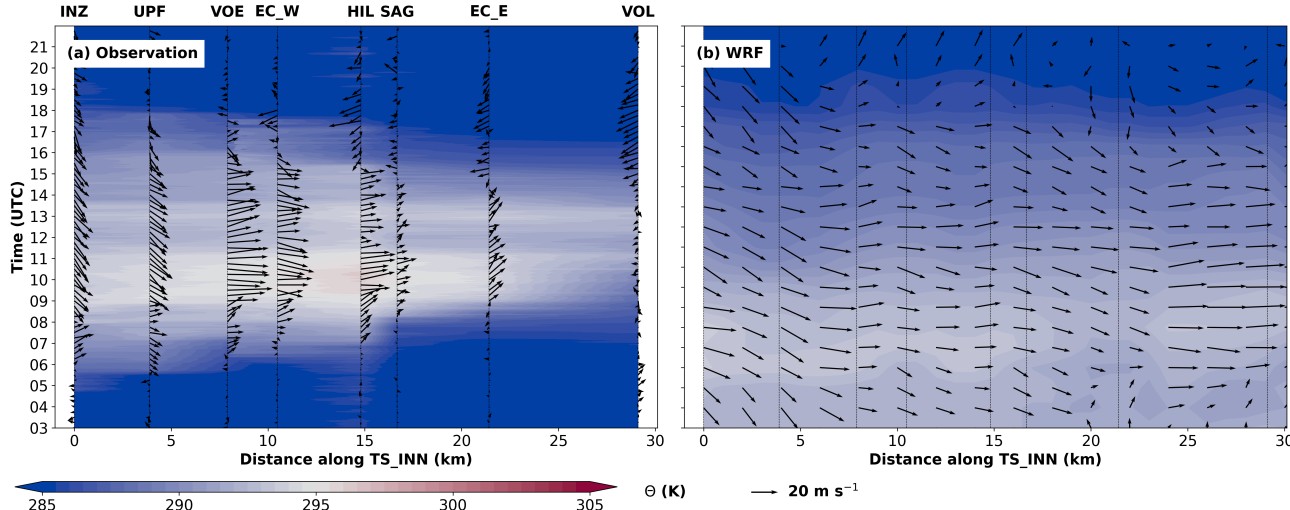

**Figure 6.** Temporal evolution of (a) observed and (b) simulated potential temperature based on Eq. (1) and horizontal wind along the valley transect TS_INN between 03 and 22 UTC 29 October 2017. Potential temperature is illustrated by color contours. Wind speed and direction are depicted by arrows. Vertical black lines indicate the location of the stations along the transect. In (a) potential temperature is shown every minute and wind arrows every 15 min. Potential temperature and wind data in (b) are based on hourly WRF output interpolated to TS_INN with wind arrows shown every second kilometer.

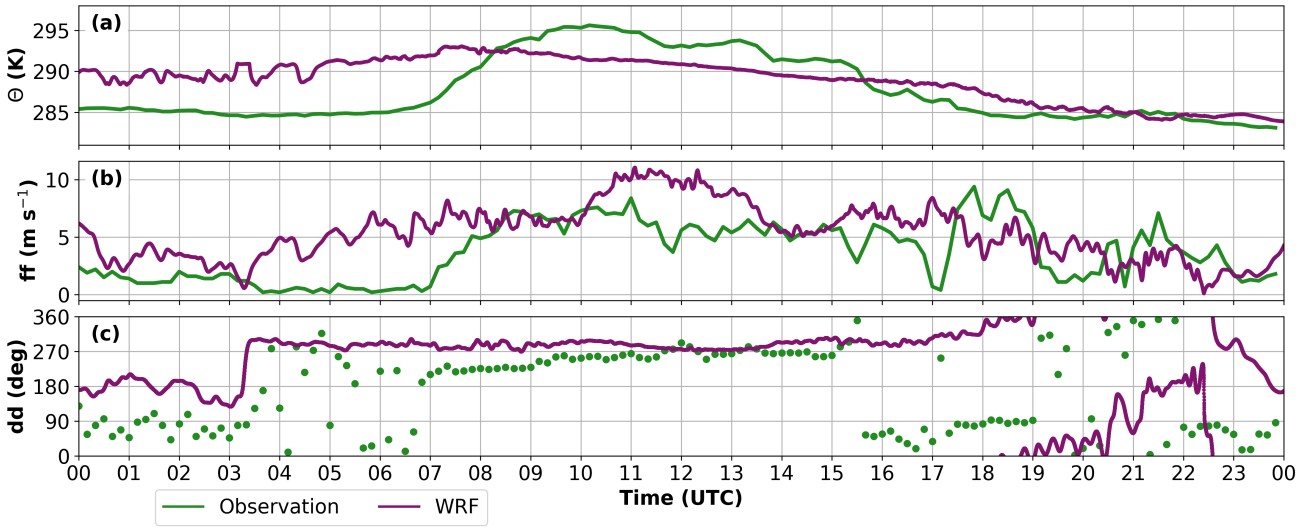

**Figure 7.** Temporal evolution of (a) potential temperature based on Eq. (1), (b) wind speed and (c) wind direction at the University of Innsbruck (UNI) on 29 October 2017. Observations are depicted in green and represent 10 min measurements. Model results are depicted in purple with values every integration time step at the WRF grid point closest to UNI.





with potentially colder air and forcing the westerlies to lift off the surface (Fig. 5a). In the later phase also flow directly form
the north could be observed in the middle of the valley atmosphere (Fig. 5a).

Figure 5 illustrates that the WRF simulation captures the overall flow pattern of the fully developed foehn in the valley
reasonably well with the northwesterly flow being channeled to a westerly wind below crest height. However, during the night
the simulated pre-foehn cold-air pool (CAP) is too shallow and not strong enough to decouple from the flow aloft (Fig. 5b).
This leads to strong westerlies inside the valley throughout the whole night (Fig. 5b) with a warm bias at the surface of about
5 K (Fig. 7a). The weak CAP stratification in the model causes a too early breakthrough of the foehn in Innsbruck. Here, the
low-level winds shift to westerlies almost four hours earlier in the model than in reality (Fig. 7c). Unlike the observations
which illustrate a down-valley progression of the foehn breakthrough, the breakthrough in the model takes place along the
valley transect TS_INN at the same time (Fig. 6b). Once the foehn is fully established in both the model and in reality the
model exhibits a cold bias of about 4 K over the whole neutral layer covering the lowest 1000 m of the valley. Wind speeds
in the model are generally higher both above crest height and within the valley (Fig. 5). In agreement with the observations,
the model depicts a local wind maximum at about 500 m above the valley surface (Fig. 5b) which is even more pronounced
at locations further upstream in the Inn Valley (not shown). The simulated foehn is strongest between about 11 and 13 UTC
both at the surface (Fig. 7b) and throughout the whole valley atmosphere (Fig. 5b). The cooling of the foehn flow in the early
afternoon is captured by the WRF simulation, as well as the strong cold air advection by the cold front. The cooling, however,
is more continuous in the simulation while in the observations it occurs in three distinct steps (Fig. 7a). Moreover, the wind
pattern during this cooling phase is different in the model than in the observations (Fig. 6 and Fig. 7b, c). This illustrates that
not only the timing but also the pathways of the cold air inflow is not perfectly captured by the model. Nevertheless, after
20 UTC the post-frontal air mass is captured well (Fig. 7a).

In summary, the WRF simulation captures large parts of the mesoscale evolution reasonably well. The regional flow patterns
as well as the precipitation pattern over the Alps are in good agreement with the observations with the exception of generally
too high wind speeds. During nighttime, the model does not correctly reproduce the cold-air pool in the Inn Valley and the
associated decoupling of the upper-level flow. However, once the foehn has fully established inside the valley, the simulated
flow patterns are in better agreement to the observations, with the exception of a cold bias of about 4 K in the model. These two
problems have been observed frequently for mesoscale simulations of foehn flows (e.g., Gohm et al., 2004; Zängl et al., 2004;
Zängl, 2006; Wilhelm, 2012; Sandner, 2020; Umek et al., 2021a). A common explanation for this is a possible misrepresen-
tation of turbulent mixing in those models. Despite these differences between the model and the observations, we will use the
simulated fields in the subsequent trajectory analysis. They should therefore not be seen as the truth in all aspects but rather as
one possible flow realization to learn more about the air mass transport form a Lagrangian perspective. Furthermore, we will
mainly focus on the fully developed foehn phase during which the model discrepancies are smaller.

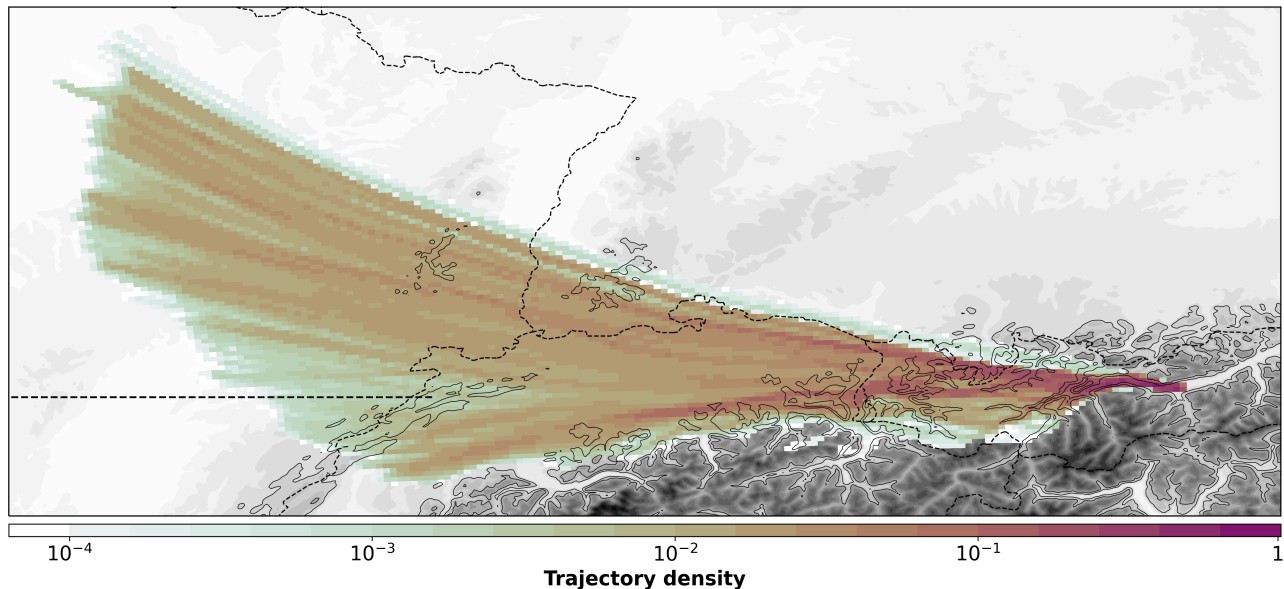

**Figure 8.** Gridded field in subdomain SD1 of trajectory density for trajectories arriving in Innsbruck at 10 UTC 29 October 2017 between 700 and 1700 m MSL. Trajectories are calculated 6 hours backward in time. Trajectory density as explained in Section 2.3.2 is indicated by color contours. Model topography is depicted outside the trajectory plume by gray shading as in Fig. 1 and inside by black contour lines of the elevations of 1000 and 1500 m MSL. National borders are shown as in Fig. 1. The latitude of 47.2° N separating the two main trajectory branches is marked by a black dashed line.

# 4 Air mass origin and history

## 4.1 Air mass origin

Based on the findings in Section 3 the following trajectory analysis concentrates on distinct phases of the foehn event. Special

emphasis is placed on the phase around 10 UTC 29 October 2017, when foehn is fully established both in the model and in reality and when model and observations are (with the aforementioned exceptions like the cold bias) in good agreement. A second focus is put on the phase around 12 UTC, when the strongest winds are simulated and when there is a change in the gravity wave pattern over the Inn Valley. To get a more complete picture, a short analysis is done for the whole foehn period between 06 and 15 UTC. In general, all analyses are restricted to trajectories reaching the valley atmosphere above Innsbruck

between 700 and 1700 m MSL. This is the well-mixed layer characterized by a nearly dry adiabatic lapse rate (Fig. 5b). Therefore all trajectories that arrive in this layer can, in principle, reach the valley floor by vertical turbulent mixing, an effect that is not explicitly taken into account in the trajectory model LAGRANTO.

Figure 8 shows that the foehn air reaching Innsbruck at 10 UTC 29 October 2017 has its origin six hours earlier distributed over a wide region in the northwestern part of the model domain. A large part of the trajectories starts over eastern France at a





**Figure 9.** Gridded fields in subdomain SD1 of various trajectory properties for trajectories as shown in Fig. 8. Depicted are the median of (a) trajectory height, (b) precipitation intensity at the surface, (c) potential temperature deviation, (d) ice-liquid potential temperature deviation, (e) water vapor mixing ratio deviation, (f) total water content deviation, (g) diabatic heating rate, and (h) adiabatic heating rate. All deviations refer to the corresponding reference value at the trajectory arrival point Innsbruck. Trajectory properties are indicated by color contours, model topography and national borders as in Fig 8, main flow branches are indicated the contour line of a trajectory density of 0.03 (thin black line). The latitude of $47.2°$ N separating the two main trajectory branches is marked by a black dashed line.





mean height of about 1200 m MSL (Fig. 9a). A second major branch representing about 25% of the trajectories (Fig. 10) origi-
nates in the northern part of Switzerland south of the mountains of the Swiss Jura, also at a mean height of about 1200 m MSL
(Fig. 9a). It is noteworthy that with 1200 m MSL the mean arrival height in Innsbruck is about the same as the mean trajectory
height upstream of the Alps. Both major flow branches funnel towards the Alps and impinge on the mountains in the area
around Lake Constance and the Rhine Valley. They then get lifted and cross the mountain ranges at the northern edge of the

Alps at heights between 2000 and 2500 m MSL and later descend into the Inn Valley. The trajectories of the southern branch
end up in the Inn Valley on average at slightly higher levels (Fig. 10a, f). As will be seen later in Section 4.2 the trajectories
of the northern branch and the ones of the southern branch experience very different diabatic processes, therefore they are
analyzed separately in the following. The separation into a northern and a southern branch is based on the trajectory starting
latitude six hours before the arrival which is either north or south of 47.2° N. This reference latitude has been chosen manually

based on the density distribution shown in Fig. 8 and the moisture fields in Figs. 9e and f. It is also very close to the latitude of
the trajectory arrival point in Innsbruck, i.e., 47.3° N. Since there is only a very low trajectory density around this latitude, the
results depend very little on the exact choice of this latitude.

The trajectories starting in France are mainly concentrated into one air stream that crosses the mountain ranges of the Vosges
and the Black Forest and reaches the Alps near Lake Constance about 1.5 hours before the arrival in Innsbruck (Fig. 8 and

Fig. 10a). The major part of these trajectories, more than 80% (see Fig. 10a), start concentrated within a narrow layer between
1000 and 1500 m MSL. However, secondary air streams originating from higher levels also play a role (Fig. 9a). The trajectories
pass the Vosges and the Black Forest mountains on a nearly terrain-following path with an average height of about 1000 m
above the local topography (Fig. 10a). Thus, the trajectories get lifted by about 200 m at the Vosges and by about 500 m at the
Black Forest. In the lee of the Vosges the trajectories experience a strong wave signal with two wave troughs parallel to the

mountain crest (Fig. 9a). A minor part of the trajectories originating from France starts further south at a slightly lower altitude
of about 1000 m MSL (Fig. 9a). Part of these trajectories do not cross the Vosges and the Black Forest, but pass them laterally
in the south and funnel into the main air stream before reaching the Alps.

About 25% of the trajectories reaching Innsbruck at 10 UTC 29 October in the lower valley atmosphere can be assigned to
the southern branch from Switzerland (blue line in Fig. 11d). This branch results from two air streams merging over the Swiss

Plateau (Fig. 8). The northern air stream subsides as a west foehn on the leeward side of the Swiss Jura (Fig 9a), while the
southern one is being channeled to a southwesterly flow between the Alps and the Swiss Jura. The merged flow is first parallel
to the northern Alpine rim and ultimately impinges on the Alps near the Rhine Valley at a mean altitude of about 1300 m MSL
about 2 hours before the arrival in Innsbruck (Fig. 10f).

## 4.2  Diabatic processes

### 4.2.1  Origin north of 47.2° N

The air along the trajectories starting north of 47.2° N experiences a mean diabatic heating of about 2.5 K in terms of potential
temperature and loses on average about 1 g kg$^{-1}$ of water vapor and about 1.3 g kg$^{-1}$ of total water content until the arrival



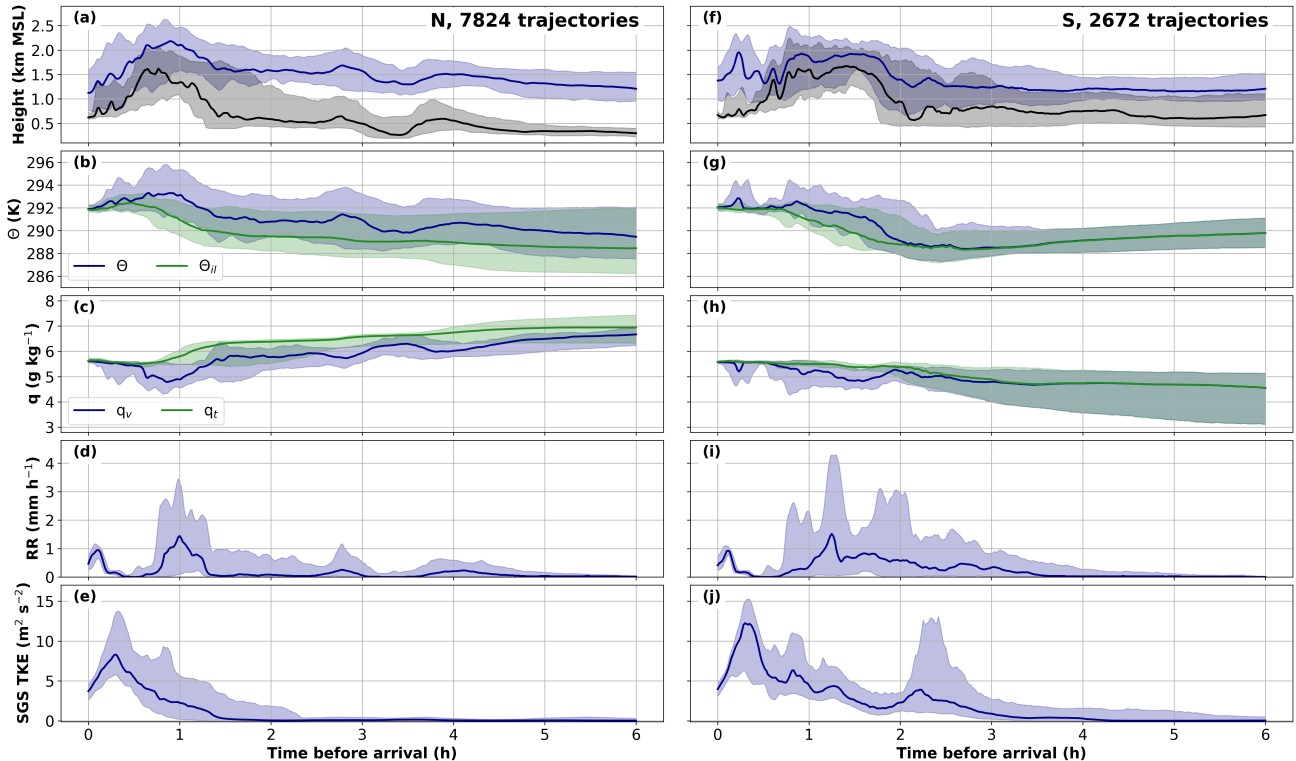

**Figure 10.** Temporal evolution of trajectory properties for trajectories as shown in Fig. 8 starting north (left column) and south of 47.2° N (right column). Thick lines indicate the median, shading the 10th and 90th percentile. (a), (f) Trajectory height (blue) and terrain height (black) in km MSL; (b), (g) $\theta$ (blue) and $\theta_{il}$ (green) in K; (c), (h) $q_v$ (blue) and $q_t$ (green) in g kg$^{-1}$; (d), (i) precipitation intensity at the surface in mm h$^{-1}$; (e), (j) subgrid-scale (SGS) TKE in m$^2$ s$^{-2}$.

in Innsbruck (Fig. 10b, c). This is especially true for the trajectories within the main air stream. Those starting at lower levels further south in France experience a much stronger warming of 5 K and a moisture loss up to 2 g kg$^{-1}$. The ones originating

from higher levels start potentially warmer and, thus, are diabatically cooled by about 3 to 5 K during their journey to Innsbruck, but only lose about 0.5 g kg$^{-1}$ of total water content (Fig. 9c, e, f). For the main air stream the strongest diabatic processes occur while crossing the mountain ranges of the Vosges, the Black Forest and the Alps. Here diabatic heating takes place when the trajectories are being lifted at the windward side while diabatic cooling is happening during subsidence on the leeward side (Fig. 9a, c and Fig. 10a, b). These diabatic temperature changes are associated with moisture changes. On the windward side

of the mountain ranges a decreasing water vapor mixing ratio points towards the formation of hydrometeors and, thus, diabatic heating due to latent heat release. The opposite is happening on the leeward side with increasing water vapor mixing ratios due to evaporation of the hydrometeors and consequently evaporative cooling (Fig. 9c, e and Fig. 10b, c). The diabatic and adiabatic heating rates are strongly anticorrelated and not only appear in the direct surrounding of the major mountain ranges,




but also exhibit patterns of trapped gravity waves upstream of the Vosges (Fig. 9g, h). This anticorrelation can be explained by
the vertical motion in these gravity waves. Updrafts cause adiabatic cooling due to expansion and, under saturated conditions,
diabatic heating due to hydrometeor formation. In the same sense downdrafts cause adiabatic heating due to compression and
diabatic cooling due to hydrometeor evaporation. It is noteworthy, however, that the adiabatic heating rates are about three
times larger than the diabatic ones (cf. Fig. 9g, h).

With the passing of all three mountain ranges a net warming in terms of potential temperature and a decrease in total water
content can be observed (Fig. 9c, f and Fig. 10b, c). Between the start of the trajectories and their arrival at the Alps a net
diabatic heating of about 1.5 K has taken place with about 0.5 K of net warming at the Vosges and 1 K at the Black Forest
(Fig. 10b). After the flow has reached the Alps a further net increase in potential temperature of about 1 K can be observed
(Fig. 10b). All three net increases in $\theta$ are accompanied by an increase in $\theta_{il}$ (Fig. 9d and Fig. 10b), which illustrates that
the heating is due to irreversible moisture loss associated with orographic precipitation (Fig. 9b, f and Fig. 10c, d). While the
net warming is approximately the same when crossing the Black Forest and the Alps, the changes in $\theta_{il}$ and $q_t$ as well as the
precipitation intensity at the surface are much stronger in the case of the Alps. At the same time, subgrid-scale TKE along the
trajectories is nearly zero over the pre-Alpine terrain, but strongly increases over the Alps up to mean values of about 8 m$^2$ s$^{-2}$
(Fig. 10e). These high TKE values suggest that turbulent mixing is a non-negligible heating/cooling source over the Alps.
However, this effect can not be quantified in this study.

### 4.2.2   Origin south of 47.2° N

Compared to the northern trajectory branch, the air along the southern branch experiences a similar net diabatic heating of
about 2 K (Fig. 9c and Fig. 10g). However, the ways in which this heating is achieved are quite different. The air along the
northern branch loses moisture along the way and large parts of the diabatic heating can be explained by latent heat release
during the formation of orographic precipitation. In contrast, the air of the southern branch gains moisture of about 1 g kg$^{-1}$
on average (Fig. 9f and Fig. 10h).

This moisture gain and the diabatic temperature changes occur at several different spots during the journey of the air masses.
During the first hours over the Swiss Plateau diabatic heating rates are almost zero (Fig. 9g). However, the comparatively dry
air parcels directly north of the Alpine rim (Fig. 9e, f) already gain about 1 g kg$^{-1}$ of moisture in the first three hours of their
journey (cf. 10th percentile in Fig. 10h). West of the Rhine Valley all trajectories experience a moistening presumably due
to precipitation formed at higher levels that partly evaporates at lower levels where the trajectories are located (Fig. 9b and
Fig. 10h, i). At the beginning, this is consistent with a decrease in $\theta_{il}$. After 2.5 h before the arrival, however, the increasing
$\theta_{il}$ in combination with the increasing moisture suggests the occurrence of compensating diabatic processes (Fig. 10g). With
the lifting at the Alps presumably local formation of hydrometeors leads to diabatic heating in terms of potential temperature
(Fig. 10f–h). During the flow over the Alps an increase in $\theta_{il}$ shows irreversible diabatic heating (Fig. 10g). However, the
near constant total water content together with the non-zero precipitation and subgrid-scale TKE (Fig. 10h–j) point towards a
complex interplay of processes that can not be fully disentangled in this study.



In summary, the foehn air reaching Innsbruck at 10 UTC 29 October 2017 experiences strong diabatic processes along the way. Depending on the origin of the air parcels, a very different history in terms of diabatic processes can be observed. Large parts of the diabatic temperature changes and moisture changes can be explained by the formation of orographic precipitation.

However, with the flow impinging on the Alps, the flow pattern gets more complex and turbulent processes as well as three-dimensional moisture exchanges appear to play an important role. However, these processes can not be disentangled and quantified with our methods. Therefore, a complete assessment of all processes is not possible within this study.

### 4.3 Temporal evolution

In the previous sections the analysis concentrated on trajectories arriving in Innsbruck at 10 UTC 29 October. This section will

analyze the evolution over the whole foehn period between 06 and 15 UTC.

Based on Fig. 11 three phases can be identified. The first phase spans from 06 to 10 UTC with a relatively constant trajectory fraction of about 75 % (25 %) originating north (south) of 47.2°N (Fig. 11d). This phase is further characterized by a mean diabatic heating of about 2 K in terms of potential temperature and a total moisture loss of about 1 g kg$^{-1}$ (Fig. 11b, c). While on average the air looses moisture, part of the air actually gains moisture (see positive values of the 90th percentile in

Fig. 11c). Throughout this phase the height of the air mass origin upstream of the Alps constantly decreases from about 1700 to 1200 m MSL, while the maximum trajectory height over the Alps stays approximately constant. It is noteworthy that at the end of the first phase the mean trajectory height upstream and at the arrival in Innsbruck are about the same.

The second phase between 11 and 13 UTC is characterized by an even higher contribution of the northern trajectory branch of up to 90 % (Fig. 11d). At the same time, the average diabatic heating and the total moisture loss sightly increase compared

to the previous phase (Fig. 11b, c). After 11 UTC nearly all trajectories experience a moisture loss as indicated by the negative 90th percentile in Fig. 11c. This is consistent with the decreasing contribution of the southern trajectory branch (Fig. 11d) and the fact that at earlier times only the southern branch was characterized by a net moisture increase (Sect. 4.2 and Fig. 9f). The maximum trajectory height slightly decreases to about 2400 m MSL at 13 UTC. At the same time, the decrease in the height of the air mass origin continues also in this second phase of the event. On average, the arrival height is now higher than the

height of the origin on the upstream side (Fig. 11a). The continuous descent of the height of origin over the course of the day is associated with a decreasing stability of the impinging air mass over the Alpine foreland (not shown). It illustrates a continuous transition from a "flow around" into a "flow over" regime which enables parcels from lower levels to directly traverse the Alps.

In the last phase between 14 and 15 UTC the contribution of the northern trajectory branch decreases strongly to about 60 % at 15 UTC. Hence, the contribution of the southern branch gains importance. On average, the height of the air mass

origin stays below 1000 m MSL while the maximum trajectory height decreases to about 2200 m MSL. The moisture loss decreases to about 0.5 g kg$^{-1}$ at 15 UTC and the diabatic temperature increase stays largely constant. However, at 15 UTC the distribution widens and a larger fraction of trajectories originates from higher levels (Fig. 11a). Those trajectories rather experience diabatic cooling than heating (see negative 10th percentile in Fig. 11b) and a net moisture uptake (see positive 90th percentile in Fig. 11c).



**Figure 11.** Temporal evolution of trajectory properties for trajectories arriving in Innsbruck from 06 to 15 UTC 29 October 2017 between 700 and 1700 m MSL. Upstream conditions are averages between five and six hours before the arrival in Innsbruck. (a) Upstream trajectory height (blue), maximum trajectory height (green), and trajectory arrival height in Innsbruck. The distribution of the arrival height in Innsbruck is indicated in red. Difference in (b) potential temperature and (c) total water content between the upstream condition and at the arrival in Innsbruck. The median in (a)–(c) is indicated by a thick line and the range between the 10th and 90th percentile as color shading. (d) Fraction of trajectories starting either north (blue) or south (green) of 47.2°N. (e) Fraction of trajectories crossing the Inn Valley boundary at four different sections as defined in Section 2.3.2.



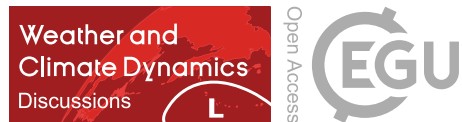

**Figure 12.** Density of trajectories crossing the Inn Valley boundary (see line IVB in Fig. 1b) through various different sections. Shown is the density on transect IVB for trajectories arriving in Innsbruck at (a) 10 and (b) 12 UTC 29 October 2017. Color contours indicate the fraction of all trajectories passing through a vertical rectangle with 1 km width and 100 m height. The histogram in the lower panel (right panel) shows the horizontal (vertical) distribution of the trajectory density summed up in vertical (horizontal) direction. The three stacked bars in the histogram correspond to the fraction of trajectories arriving in Innsbruck in the three layers 700 to 1200 m MSL, 1200 to 1700 m MSL, and 1700 to 2500 m MSL, respectively (see legend). Four subsections of the transect as described in Section 2.3.2 are indicated by horizontal color bars and are separated by vertical black lines. Important terrain features are named in (a).





## 5 Local aspects

### 5.1 Valley-boundary crossing

With more than 80 % the major part of the trajectories arriving in Innsbruck below 1700 m MSL at 10 UTC crosses the Inn Valley boundary (IVB) through section II (Fig. 11e). Most of this crossing takes place in a 20 km broad section of the northeastern Lechtal Alps, where the crestline is about 400 m lower than in the southwestern part, which facilitates crestline crossing (Fig. 12a). All trajectories pass this section within a shallow layer of about 500 m above terrain height.

A minor portion of trajectories enters the Inn Valley further west in the region of the Arlberg and Silvretta Pass via the Stanzer and Paznaun Valley through section I (Fig. 12a). Their contribution is only about 10 % at 10 UTC (Fig. 11e) Trajectories entering this way are already below the surrounding crest height as they cross the valley boundary (Fig. 12a). For earlier times some trajectories even enter the Inn Valley through the Lower Engadine (not shown). It is noteworthy that the southern trajectory branch originating from Switzerland has a comparatively large contribution of about 40 % to trajectories entering via section I (not shown).

A tiny part of less than 1 % crosses the Inn Valley boundary at 10 UTC through section III (Fig. 11e), which represents the Wetterstein Mountains and the Mieming Chain (Fig. 12a). A negligible part passes through the most western part of Section IV which represents the Erlspitze Group of the Karwendel. The crossing takes place in a narrow filament located between about 2500 and 3500 m MSL (Fig. 12a).

At 10 UTC, the four sections contribute differently to the distribution of trajectory arrival heights. Trajectories crossing the Lechtal Alps within the main cluster end up distributed over the whole valley atmosphere (lower histogram in Fig. 12a). In contrast, most trajectories passing through section I reach Innsbruck mainly in the mid and upper valley atmosphere. Trajectories crossing through the aforementioned filament in section III and IV only reach Innsbruck in the upper valley atmosphere above 1700 m MSL. In summary, only trajectories passing the Lechtal Alps end up in the lower valley atmosphere at this time.

After 10 UTC a strong change in the patterns of the valley boundary crossing can be identified (Fig. 11e). Between 10 and 12 UTC the contribution through section III (Wetterstein Mountains and Mieming Chain) increases from zero to more than 80 %, while the contribution through section II (Lechtal Alps) decreases to less than 20 %. After this pattern change almost no trajectory enters through section I, while the contribution through section IV (Karwendel) increases from zero to a few percent at the very end of the event (Fig. 11e).

This shift in pattern for the trajectories arriving at 10 and 12 UTC can be seen by comparing Fig. 12a and Fig. 12b. The majority of trajectories entering at 12 UTC through section II still cross the valley boundary mainly in the northeastern part of the Lechtal Alps below 2500 m MSL and still arrive in Innsbruck distributed over the whole valley atmosphere. However, the crossing through section III dominates now (Fig. 12b). The terrain-following structure of the trajectory plume in section III and IV is similar to that at 10 UTC. At this time, however, the plume is located at a slightly lower height and has a broader vertical extent (Fig. 12b). For the trajectories entering through the sections III and IV a strong dependency between the crossing position, the crossing height and the arrival height can be observed (Fig. 12b). Here the distributions of the crossing positions





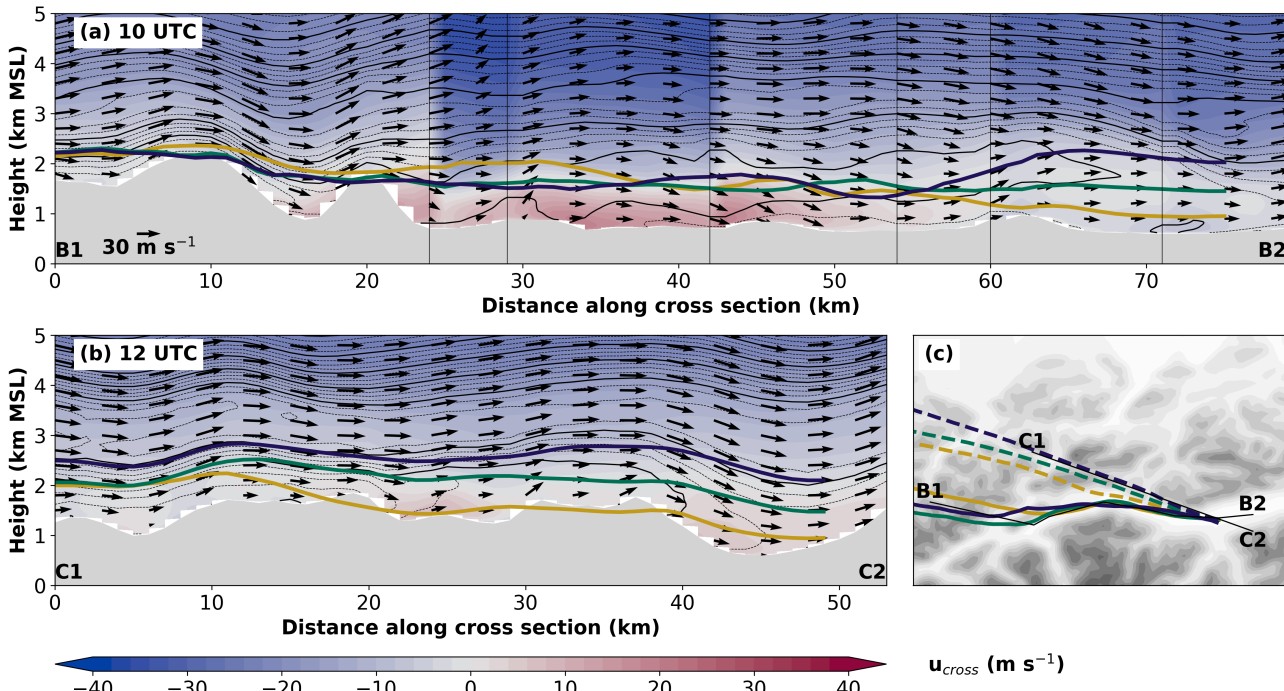

**Figure 13.** Vertical transect of equivalent potential temperature and wind along (a) CS_LTA at 10 UTC and (b) CS_WST at 12 UTC 29 October 2017 and median trajectory height selected by the arrival height in Innsbruck and the dominant section where the trajectories cross the Inn Valley boundary. Black contour lines and wind vectors as in Fig. 5. Color contours represent horizontal wind component perpendicular to the transect (positive into the plain). Discontinuities in (a) are caused by kinks in the transect (see alignment in (c)). Lines indicate median trajectory height projected to the transect for trajectories crossing section II in (a) and section III in (b). Line colors indicate the trajectory arrival heights as in Fig. 12. Topographic map in (c) with median trajectory position for trajectories arriving via section II at 10 UTC (thick solid lines) and section III at 12 UTC (thick dashed lines). Thin black lines indicate the location of the cross sections, and text labels illustrate the start and end points of the transects.

for the trajectories arriving in Innsbruck at higher levels are shifted towards positions further east and towards higher levels on the valley boundary line IVB.

## 485  5.2  Penetration mechanisms

In the previous section a drastic temporal change was identified in where and how the airflow passes the Inn Valley boundary. In the second half of the event the major part of the trajectories enter the Inn Valley closer to Innsbruck. The main objective of the following sections is to explore the dominating penetration mechanisms and the diabatic processes on the leeward side before and after that pattern shift.





### 5.2.1 Lechtal Alps

Until 10 UTC the vast majority of trajectories enters the Inn Valley via section II over the Lechtal Alps (Fig. 11e and Fig. 12a). These trajectories stay close to the terrain at about 2000 m MSL before they enter the Inn Valley (Fig. 13a). They are trapped near crest height in an about 500 m deep layer of high stability. In this phase trajectories of different arrival heights stay initially close together over the high terrain. In the lee of the Lechtal Alps the trajectories descend in a gravity wave below crest height

(Fig. 13a) and get deflected and channeled by the Inn Valley (Fig. 13c). Some of the trajectories already descend down to the valley floor directly on the northern slope of the Inn Valley (not shown), but the majority stays in the upper valley atmosphere just below crest height.

    As seen in Sect. 3 a rotor covering the whole valley atmosphere is dominating the flow field in this region of the Inn Valley. This can also be seen in Fig. 13a between $x = 25$ and $55$ km by a local minimum of $\theta_e$ at heights around 1500 m MSL and

strong cross-valley winds with positive (negative) values below (above) 1500 m MSL. As the cross-valley rotor circulation is superimposed by a westerly along-valley flow (see Fig. 3 and along-valley component in Fig. 13a), this results in a helix-shaped flow pattern which leads to the helix shape of the three different mean trajectories in Fig. 13a. Until the region of Zirl (see Fig. 3 and $x = 55$ km in Fig. 13a) the trajectories are located within this rotor with the median trajectory height for all arrival heights located in the center of the rotor at about 1500 m MSL. In the region of Zirl, where the Inn Valley changes its orientation, the

trajectories of different arrival heights are quickly separated vertically from each other. This could at least partly be caused by trajectories impinging on the southern foothills of the Karwendel (see hill in Fig. 13a at around $x = 63$ km) and by a hydraulic jump in the lee of the northern Stubai Alps south of the Inn Valley (not shown).

### 5.2.2 Wetterstein, Mieming Chain, and Karwendel

  After 10 UTC a shift in the penetration pattern takes place with the majority of trajectories now crossing the valley boundary

in section III (Fig. 11e). However, especially for the trajectories arriving at Innsbruck in the mid and upper valley atmosphere, a clear distinction between a penetration in the lee of the Wetterstein and Mieming Chain (section III) and the Karwendel (section IV) is not possible as the trajectories flow almost parallel to the line IVB in this region (cf. Fig. 1b and Fig. 13c). As shown in Fig. 13b, these trajectories pass first the Wetterstein and Mieming Chain on average between 2000 and 2500 m MSL. Afterwards they stay above crest height nearly parallel to the crest line (Fig. 13c) until they finally penetrate into the Inn Valley

in the lee of the most western part of the Karwendel (Erlspitze Group). In other words, trajectories first pass section III and afterwards part of them also section IV. This can not be seen in Fig. 12b which only illustrates the location where the trajectories pass IVB first. In contrast, the trajectories reaching Innsbruck in the lower valley atmosphere subside below crest height in the lee of the Mieming Chain but remain on average in the mid valley atmosphere until their final penetration to the valley floor right before arriving at Innsbruck (Fig. 13b). In both cases, the final penetration into the valley is caused by subsidence in a

gravity wave. It is noteworthy that in contrast to the situation at 10 UTC a clear horizontal and vertical separation between the trajectories arriving at different heights in Innsbruck already occurs with the crossing of the Inn Valley boundary.





**Figure 14.** Vertical transect of the wind field and the median trajectory height as in Fig. 13 for (a) CS_LTA at 10 UTC and (e) CS_WST at 12 UTC 29 October 2017. The black contour line indicates a relative humidity of 99%. Color contours represent the total hydrometeor mixing ratio $q_t - q_v$. Panels (b)–(d) and (f)–(h) show median trajectory properties for the three arrival heights (same line colors as in Fig. 12). (b), (f) $\theta$ (solid line) and $\theta_{il}$ (dashed line); (c), (g) $q_v$ (solid line) and $q_t$ (dashed line); (d), (h) relative humidity.





## 5.3 Diabatic processes on the leeward side

### 5.3.1 Lechtal Alps

The trajectories arriving in Innsbruck at 10 UTC pass the Lechtal Alps saturated within the clouds (Fig. 14a, d). Here lifting
at the ridge of the Lechtal Alps leads to further formation of hydrometeors and thus to temporary diabatic heating (see low
arriving trajectories, Fig. 14a–c), while the continuous increase in $\theta_{il}$ illustrates an irreversible warming due to orographic
precipitation and the associated moisture loss (Fig. 14b, c). Subsidence in the lee of the Lechtal Alps finally leads to drying
of the air and diabatic cooling due to the evaporation of hydrometeors (Fig. 14a–d). Once the trajectories are deflected into
the valley direction, they stay within the rotor circulation. Here, the trajectories remain on average in the slightly subsaturated
air below the cloud base (Fig. 14a, d). After the vertical separation of the trajectories at about 15 km before the arrival in
Innsbruck, only the fraction of the trajectories arriving in the upper valley atmosphere gets lifted into the cloud again, leading
to diabatic warming and the associated formation of hydrometeors and subsequent moisture loss.

### 5.3.2 Wetterstein, Mieming Chain, and Karwendel

The trajectories arriving in Innsbruck at 12 UTC also pass the mountians within the clouds (Fig. 14e). Here, lifting upstream
of the Wetterstein and Mieming Chain leads to the formation of more hydrometeors and consequently to diabatic heating
(Fig. 14e–g at about $x = 10$ km). The same process is taking place for the trajectories arriving in the upper valley atmosphere
upstream of the Karwendel ($x = 30$ km in Fig. 14e–g). The low-arriving trajectories descend below the cloud base already
with their subsidence to the mid valley atmosphere in the lee of the Mieming Chain and stay subsaturated until their arrival in
Innsbruck (Fig. 14e, h after about $x = 15$ km). In contrast, the trajectories arriving at the upper levels stay in the clouds all the
time, while the ones arriving at the mid levels become subsaturated just in the lee of the Karwendel about 5 km before arriving
in Innsbruck (Fig. 14e, h). Here subsidence in the gravity wave leads to drying while parts of the hyrdometeors get advected
into subsaturated air (Fig. 14e). Within the clouds, moisture loss due to precipitation falling out (Fig. 14g) leads to continuous
irreversible diabatic heating, illustrated by the increase in $\theta_{il}$ in Fig. 14f.

## 6 Discussion

### 6.1 Penetration into the Inn Valley

Our results partly confirm previous findings on north foehn in the Inn Valley. In agreement to Zängl (2006), the penetration into
the Inn Valley took place via several different pathways covering the entire region west of Innsbruck. Zängl (2006) suggested an
increasing contribution of the Arlberg region with a stronger westerly component of the large-scale flow, or with the presence
of a barrier jet in the Alpine foreland. This partly agrees with the high contribution of the southern air stream from Switzerland
to the trajectories penetrating via section I in this study. However, a simple direct relation does not seem to exist, given the fact
that the contribution of the trajectories form Switzerland increased towards the end of the foehn event although almost none of





them passed through section I (cf Fig. 11d and Fig. 11e). In other words, the trajectory origin does not necessarily determine the region where the trajectories enter the Inn Valley.

In agreement with Zängl (2006), we identified gravity waves in the lee of the Lechtal Alps as one important mechanism for the air flow to penetrate into the Inn Valley. However, we also found an additional mechanism that has not been mentioned before which is a persistent cross-valley rotor circulation in the Inn Valley. The penetration associated with this rotor resembles scenario D in Strauss et al. (2016) with a large atmospheric rotor covering the whole valley atmosphere underneath a trapped lee wave. The subsidence on the downstream side of the rotor (which is the windward side of the subsequent mountain ridge) provides an effective mechanism to import air into the valley atmosphere. The decreasing stability and increasing wind speed above 2500 m MSL (Fig. 4a) result in a vertically decreasing Scorer-Parameter (Scorer, 1949) and, thus, provide an environment favorable for the formation of such trapped lee waves (Lin, 2007). The rotor therefore resembles a type I rotor according to Hertenstein and Kuettner (2005). Downstream topography has been shown to impact the lee wave pattern through wave interference (Grubišić and Stiperski, 2009; Stiperski and Grubišić, 2011). Such downstream topography is in our case, e.g., the mountain range of the Stubai Alps south of the Inn Valley as well as the ridge between Tschirgant and Simmering northwest of Haiming (see Fig. 3 and hill around $x = 20$ km in Fig. 13a). Given the fact that no rotor formed in the slightly wider part of the Inn Valley east of Innsbruck (Fig. 3, Fig. 13b) the rotor formation appears to be strongly dependent on the valley geometry.

Similar to Strauss et al. (2016), the position and strength of the rotor depended on the associated gravity wave field aloft. From 10 to 12 UTC the horizontal wavelength and the amplitude of the trapped lee wave increased (Fig. 4b). This resulted in stronger subsidence on the northern slope of the Inn Valley and a downstream shift of the rotor position which weakened the rotor-induced subsidence and the associated penetration mechanism. A possible reason for this change in the wave pattern could be the weakening of the crest-level stability (which is consistent with the weakening of the upstream stability discussed in Section 4.3) and a slight increase in crest-level wind speed (see Fig. 4b and Fig. 5b, around 2500 m MSL). In Xue et al. (2020) a weakening of the upstream stratification together with reduced upstream blocking, and a strengthening of the cross-mountain winds led to a widening and an amplification of the lee wave. Similarly, in Strauss et al. (2016) increased crest-level wind speeds led to an increase of the horizontal wavelength and the wave amplitude, which resulted in a downstream shift of the rotor position and strongest wind speeds at the valley floor. Apart from the changes in the upstream stratification, a slight shift in the crest-level wind direction may have led to changes in the effective distance to the downstream topography resulting in a changed interference pattern (Grubišić and Stiperski, 2009; Stiperski and Grubišić, 2011). Additionally this shift to a slightly stronger northerly wind component presumably facilitated the penetration in the lee of the Wetterstein and Mieming Chain given the east-west orientation of the Inn Valley in this area. Hence, the lee wave pattern and, thus also the rotor and its associated penetration pattern, is very sensitive to subtle changes in the upstream conditions.

Our results disagree with those of Zängl (2006) for the air flow penetrating into the Inn Valley in the lee of the most western part of the Karwendel in the second phase of the event. Zängl (2006) argued that in his case the northerly flow in the lee of this mountain ridge resulted from deflection of the downvalley flow by a local terrain corrugation. However, in our case the trajectories passing the Karwendel did not originate from inside but rather from far north of the Inn Valley (Fig. 13c). Moreover, the final penetration of air into the valley was partly caused by a gravity wave in the lee of the Karwendel which was not present





in the case of Zängl (2006). The pathway through the terrain gap of Seefeld as suggested in Haas (2006) did not play a role for the air mass penetration in our case. Our results illustrate that the locations and mechanisms of airflow penetration are strongly time- and case-dependent. Furthermore, they can only be convincingly determined by combining Lagrangian and Eulerian
perspectives.

In contrast to Zängl (2006), orographic precipitation did not suppress the evolution and breakthrough of foehn in the Inn Valley in our case. Our results are closer to those of Haas (2006), with precipitation being formed upstream and being advected into the Inn Valley, resembling the characteristics of "dimmerfoehn" (Richter and Hächler, 2013). Zängl (2006) reasoned that precipitation falling into layers of unsaturated air in the valley stabilized these layers due to evaporative cooling, leading to
a dampening of the amplitude of gravity waves and thus preventing the foehn from penetrating. In our case the air along the trajectories experienced a net diabatic warming while crossing the Alps due to the formation of orographic precipitation. Inside the valley the evaporative cooling from precipitation formed above the air parcel only played a marginal role and the air inside the valley stayed largely well-mixed.

## 6.2   Foehn classification

Due to the channeling of the flow into the valley direction most foehn-like flows originating north of the Alps appear as strong westerly to northwesterly winds in Innsbruck, independent of where the penetration into the valley takes place. Because of that, different authors have used different definitions, so that there exists no uniform classification to distinguish between west foehn, northwest foehn, and north foehn. Following the local-scale approaches based on the wind direction in Innsbruck of Trabert (1903) and Wankmüller (1995), the case presented here would have to be called a west foehn (see Fig. 6 and Fig. 7). In
Fig. 6, however, we already can see the weakness of this approach, as the wind direction varies strongly along the valley due to changes in the valley orientation. Strictly following their approach, this would result in different classifications for the same event at different locations. Following the larger-scale approaches of Zängl (2006) and Haas (2006), the case would have to be called a north foehn. Arguments for this are the large-scale flow crossing the Alps with a northerly component above crest height and northerly flow in the Wipp Valley.

However, the trajectory analysis conducted in this study provide the possibility for two new classification approaches based on a Lagrangian perspective on (1) the large-scale air mass origin and (2) the point of penetration into the Inn Valley. With the first approach, which is an Alpine-scale perspective, one finds that the major part of the trajectories originates north of Innsbruck (Fig. 11d), more specifically in the northwestern quadrant (Fig. 8). A clear limitation of this approach is the subjectivity of the choices how long the trajectories are calculated backwards in time and how the individual sectors are defined. For the
second approach, which is a valley-scale perspective, the valley boundary and individual section have to be defined. In our case, section I has a south–north orientation and section II a southwest–northeast orientation. Hence, even for a pure westerly flow not only section I but also section II could be crossed leading to a possible penetration through these sections. Hence, flow conditions that lead to a dominant flow through section I and II could be classified as west foehn from a valley-scale perspective. Likewise, a dominant flow through section III could be called northwest foehn and a dominant flow through section IV north
foehn. A limitation of this approach is that this is only valid for Innsbruck as trajectories released from different locations along





the Inn Valley will presumably experience different contributions. Based on these two approaches the foehn in this study could be classified as a *northwest* foehn on an Alpine scale throughout the whole event, while on the valley scale a gradual transition from a *west* foehn to a *northwest* foehn is more appropriate.

Apart form the specific limitations of the two new approaches presented above, a trajectory-based classification would have other more general limitations. Turbulent processes may not be sufficiently captured by the model and are not represented in the trajectory calculation resulting most likely in an underestimation of the trajectory dispersion. Furthermore, a trajectory analysis is typically not available on an operational basis, which excludes its use for real-time foehn diagnosis and forecasting. Last but not least, a "binary" classification in one or the other category remains subjective and is not able to reflect the variety of penetration mechanisms and their transient contributions.

## 7 Conclusions

In this study the foehn event of PIANO IOP 1 on 29 October 2017 was investigated based on a mesoscale numerical simulation, backward trajectories and observations. The aim was to determine the origin, the pathways and the entrance regions of the air mass penetrating into the Inn Valley and finally arriving in Innsbruck. Furthermore, diabatic air mass transformation processes from the windward to the leeward side over various mountain ridges were assessed. The most important results are:

- The main phase of the foehn event lasted from the morning to the afternoon of 29 October 2017 with the foehn break-through at the valley floor in Innsbruck at about 08 UTC and the foehn breakdown at 1530 UTC when a cold front arrived. During the foehn period the strong northwesterly flow above crest height was channeled into the valley direction and appeared as a strong westerly wind in Innsbruck. Orographic precipitation occurred especially on the northern slopes of the Alps, but was also observed in the Inn Valley.

- On the Alpine scale the WRF simulation was able to reproduce the regional flow patterns of low-level deflection and upper-level crossing as well as the pattern of orographic precipitation. On the valley scale the model underestimated the strength of the nighttime cold pool and the associated decoupling of the valley atmosphere from the flow aloft. Consequently, the flow in the valley was too strong and the foehn breakthrough at the valley floor occurred too early. However, once the foehn was fully established also in reality, the simulated flow field agreed reasonably well with the observations, apart form a cold model bias of 3 to 4 K in the lowest 1000 m AGL.

- Backward trajectories calculated over the duration of six hours revealed the location of the air mass origin in the north-west of the Inn Valley with two pronounced air steams. The main (northern) air stream originated in France, crossed the mountain ranges of the Vosges and the Black Forest and finally reached the Alps in the region of Lake Constance and the Rhine Valley. This air stream contributed with 60 to 90 % to the foehn air arriving in Innsbruck. The secondary (southern) air stream originated over the Swiss Plateau and was aligned parallel to the Alpine rim until it finally impinged on the Alps in the same region as the main branch. During the event the contributions of the two air streams slightly changed with a maximum contribution from the northern branch of about 90 % between 11 and 13 UTC.





- For the trajectories of the main air stream, orographic precipitation over the Vosges, the Black Forest and the Alps led to moisture loss of about 1 g kg$^{-1}$ and net diabatic heating of about 2.5 K. In contrast, trajectories of the southern
flow branch experienced moisture uptake and diabatic cooling on the windward (west) side of the Alps most likely due to evaporation of precipitation from aloft. Nevertheless, subsequent diabatic heating over the Alps dominated the total heating also for the southern branch and led to a net increase in potential temperature of about 2 K along the trajectories. This increase is most likely attributed to orographic precipitation over the Alps. However, the role of other heating processes such as turbulence and the role of three-dimensional moisture exchange could not be conclusively determined.

- The entrance region, where the main part of the air mass penetrated into the Inn Valley, changed over the course of the event. Until 10 UTC the majority of the trajectories (80 to 90 %) entered the Inn Valley via the Lechtal Alps, while the remaining part largely entered the Inn Valley further to the west in the Arlberg region. After 12 UTC the dominant part (70 to 85 %) entered via the Wetterstein and Mieming Chain. A minor part entered via the most western part of the Karwendel. Penetration of air into the Inn Valley directly from the north of Innsbruck via the Nordkette was negligible at
all times. From a Lagrangian-based valley-scale perspective the shift in the penetration pattern between 10 and 12 UTC can be interpreted as a transition from west to northwest foehn.

- For the trajectories entering the valley via the Lechtal Alps two penetration mechanisms were important. A gravity wave in the lee of the Lechtal Alps forced the trajectories below crest height and for a smaller fraction allowed a penetration all the way to the valley floor. For the majority of trajectories, however, a rotor underneath this lee wave stretching over
a distance of about 30 km played a crucial role in bringing the trajectories into the valley atmosphere.

- For the trajectories entering the valley closer to Innsbruck the penetration was associated with a gravity wave in the lee of the Wetterstein and Mieming Chain, as well as in the lee of the most western part of the Karwendel.

- A weakening of the upstream stability and an increase in crest-level wind speeds together with a slight shift to a stronger northerly component likely led to a widening and an amplification of the lee waves over the Inn Valley in the second
phase of the event. This resulted in stronger subsidence on the northern slopes of the Inn Valley and a weakening of the rotor-induced penetration. Apart from that, the shift in wind direction likely facilitated the penetration in the lee of the Wetterstein and Mieming Chain, given the change in the orientation of the Inn Valley in this region.

The study showed the important role of diabatic processes on the temperature changes along the trajectories. With the methods used here, however, only a qualitative assessment of individual processes was possible. This gap could be filled by
retrieving and analyzing individual temperature tendencies from individual microphysical, turbulent and radiative processes along the trajectories in order to close the budgets of heat and moisture. This could be achieved by either using online trajectories, calculated already within the numerical model (e.g., Miltenberger et al., 2016) or by retrieving the tendency terms (e.g., Göbel et al., 2021; Umek et al., 2021a) and using them in offline trajectories similar to the investigations done here. One limitation of this study is the neglect of turbulent effects on trajectory dispersion. This could be improved by either substantially
increasing model resolution or by using trajectory tools that at least account for parameterized turbulence. However, which of

these approaches is more practical and fruitful remains to be shown. Furthermore, it has to be shown to what extent a higher model grid resolution for a better representation of stratification and turbulence in the valley atmosphere (Umek et al., 2021b) also improves the trajectory dispersion in the mountain boundary layer.

*Code and data availability.* The WRF model (https://wrf-model.org/, University Corporation for Atmospheric Research) and the trajectory
analysis tool LAGRANTO (https://lagranto.ethc.ch/, Atmospheric Dynamics Group, Institute for Atmospheric and Climate Science, ETH Zurich) are publicly available. The PIANO data set is partly available on Zenodo data repositories. The Doppler wind lidar data is available at https://doi.org/10.5281/zenodo.4674773 (Gohm et al., 2021a), the AWS data is partly available at https://doi.org/10.5281/zenodo.4745957 (Gohm et al., 2021c) and at https://doi.org/10.5281/zenodo.4672313 (Gohm et al., 2021b). Further observational data and codes for analysis are available on request from the authors.

*Author contributions.* MS performed the simulations and the analysis and wrote the manuscript as part of his Master's thesis project. AG was his supervisor and provided the idea for this study. All authors contributed equally to the interpretation of results and the finalization of the manuscript.

*Competing interests.* The authors declare that they have no conflict of interest.

*Acknowledgements.* This study was conducted in the framework of the PIANO project which was supported by the Austrian Science Fund
(FWF) and the Weiss Science Foundation under Grant P29746-N32. The PIANO field campaign was supported by KIT IMK-IFU, Austro Control GmbH, Zentralanstalt für Meteorologie und Geodynamik (ZAMG), Innsbrucker Kommunalbetriebe AG (IKB), Bergisel Betriebsgesellschaft m.b.H., Innsbrucker Nordkettenbahnen Betriebs GmbH, T-Mobile Austria GmbH, Unser Lagerhaus Warenhandelsgesellschaft, PEMA Immobilien GmbH, HTL Anichstrasse, Hilton Innsbruck, TINETZ-Tiroler Netze GmbH, Land Tirol, and the communities Patsch and Völs. The computational results presented here have been achieved (in part) using the LEO HPC infrastructure of the University of
Innsbruck.



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
