# Peer review of "Is it north or west foehn? A Lagrangian analysis of PIANO IOP 1"

_Weather and Climate Dynamics, 2021_

## Referee Comment (RC2)

**Review for**

**'Is it north or west foehn? A Lagrangian analysis of PIANO IOP 1' https://doi.org/10.5194/wcd-2021-65**

**Synthesis:**

The authors present a case study of a west to northwest foehn event in the Inn Valley that occurred during the PIANO field campaign in Oct 2017. The mesoscale WRF simulation is validated using the extensive measurement dataset gathered during IOP1, and then combined with Lagrangian air parcel trajectories to assess three core topics:

- Air mass origin, air streams and diabatic processes upstream of the Alps
- Inn Valley boundary crossing and the associated penetration mechanisms
- The classification of such events into north or west foehn

The study is well-written and structured in a meaningful way. However, upon reading the title of the manuscript, I wonder whether it actually addresses the most important result of the study. In my opinion, Section 5.1 and 5.2 concerning the valley boundary crossing and the associated penetration mechanisms are an innovative assessment of Foehn descent mechanisms and pathways, combining the Lagrangian and the Eulerian perspective. The authors might consider this to be reflected in the title of the study as well. However, it is purely a suggestion and the importance of the key results can, of course, be assessed differently. Besides this comment, I have some concerns regarding the verbosity of the manuscript. For example, while the large-scale overview and the WRF model validation is important to set the scene for the case study, it could be written in a more concise manner and, more directly, guide the reader towards the main conclusions of the study. Hence, I added some suggestions in the minor comments below, where I find the authors might consider re-writing and shortening certain text passages. In addition, I think the current embedment of diabatic processes into the study should be reconsidered. First of all, the respective research question could be rephrased more precisely. Secondly, the relevance of Section 4.2 and 5.3 remain, to a certain extent, unclear. Either, the authors emphasize the relevance of the discussion of diabatic processes for the penetration of the Foehn air, or embed the results into other literature addressing diabatic processes of Foehn flows. If so, I suggest adding an additional discussion chapter with respect to this topic. Or, alternatively, the results concerning diabatic processes could also be discarded from the study. I included the respective comments in the points below where I saw fit. Besides these minor concerns and some additional minor comments, I didn't identify any major concerns. Therefore, after the authors have addressed my comments, I would recommend the study for publication in Weather and Climate Dynamics.

**Minor comments:**

**Abstract:**

• L10, L18: Terminology: When invoking the Lagrangian perspective, I would either always use the phrase "air masses", "air parcels", or simply "air", but not use them as synonyms, in order to avoid terminological confusion. The authors might want to

double check this throughout the manuscript and also clearly differentiate between "air parcels" and "air stream" (the latter being a coherent bundle of air parcels).

• L24-25: It is unclear to me what kind of foehn criteria you refer to here. Is north and west foehn distinguished based on an estimate of adiabatic heating? You could specify this with additional information, or re-write the last sentence of the abstract.

**Introduction:**

- Fig. 1: If I checked it correctly, the abbreviation IVB (Inn Valley boundary) is not defined where it is first used.
- L77: "... various diabatic and adiabatic processes within the flow" sounds a bit strange to me. Maybe you could leave "within the flow" or replace it by "along the transport pathway".
- L78: "this"  $\rightarrow$  "these"
- L79-L80: Could you specify a bit more what kind of events you refer to? "heat events" are probably "heat events in Japan", "melt events" are "melt events in polar regions"?
- L84-85: It seems a bit unclear to what location the 20% refer to. To Innsbruck (i.e., the Inn Valley itself)? Or does this refer to upstream precipitation frequency? Please rephrase accordingly.
- L91-105: As one suggestion to shorten the manuscript: Is this discussion of the impact of moisture on gravity waves really relevant, or could it be shortened?
- L115: Research question three could be formulated in a more precise manner. It remains unclear, whether the authors refer to the role of adiabatic and diabatic processes for the descent or the heat budget of air parcels. You might want to consider rephrasing the third goal of the study (see also related comment in the synthesis).

**Data and methods:**

- Section 2.1 and Section 2.2 are both well-written, but rather extensive. Perhaps, the authors find one to two passages where they could shorten the methods section a bit, e.g., the listing of the measurements and stations used (L123-138), or the detailed description of the model setup (L160-169).
- L188: I don't understand how the total number of trajectories can be only 18'688. If you have dx=dy=200 m on a 4x4 km square, this would result in 21 x 21 = 441 trajectories per horizontal plane. Combining it with dz = 25 m over a distance of 2500m 700m = 1800m, I end up with 441\*73 = 32'193 trajectories. Maybe I misunderstood the trajectory setup. In that case, I would be glad if you can clarify it shortly.
- L191-192: Could you please explain what this jump option implies?
- L200: Please give some references to these other trajectory tools.
- L202-203: I think I understand what you mean but the sentence seems a bit hard to understand at first. Maybe rephrase in an easier way.
- L210: Are there not five hydrometeor classes in the microphysics scheme (cloud water, rain, cloud ice, snow, graupel)?
- L227: Is there any residual resulting from the equation applied to your data? Could you please comment on its magnitude (either here or when you present the respective results in Section 4.2)?

• L235-L240: I had difficulties in understanding this entire passage. I think you should consider reformulating the description of the Eulerian mapping of Lagrangian trajectory information. Also, I'm unsure whether you calculate "mean trajectory properties" or "median properties", since you write "by taking the median of various parameters".

**Meteorological overview:**

- Section 3 is, as well, rather extensive considering it is not addressing any research question directly. However, the meteorological overview is still helpful to understand the case study and, therefore, it is fine to have an extensive overview. However, I added one suggestion to shorten a passage below.
- Fig. 5: With the chosen colorbar I have some difficulties inferring on the vertical stratification from the figure. Maybe a few contour lines of potential temperature could be added for readability?
- L269: Since the flow deflection is not visible very clearly at 700 hPa, you could add "(not shown)" to the first part of the sentence. Or, alternatively, display the 850 hPa wind field in Fig. 2b.
- Fig. 7 caption: The phrasing "... with values every integration time step at the WRF grid point closest to UNI" confuses me a bit. Probably you rather mean "... with values from every output time step of the WRF grid point closest to UNI."?
- L298-315: Could this be more concise since the comparison just serves for the model validation?
- L296: typo ("form" → "from"). I saw this typo multiple times, please check at other places throughout the manuscript.
- L318: "cold-air pool" → "CAP"
- L324-325: The meaning of this sentence is a bit hard to grasp. Maybe you could rephrase it or explain it a bit more precisely.

**Air mass origin and history**

- Section 4.2 As mentioned in the synthesis, the goal of this Section remains a bit unclear What do we learn from it? Is it relevant for the penetration into the Inn Valley? In my opinion, it would need a bit more context why the thermodynamic assessment is of relevance for the foehn community (e.g., in an additional, short discussion chapter).
- Section 4.3: The Section is very interesting, since it aims to explain the differing thermodynamic properties and altitude of origins by the varying importance of two different air streams. However, the reader is left with two open questions at this point: First of all, since the total moisture change, the altitudes and the total potential temperature change is shown for all trajectories, their temporal evolution can only indirectly be linked to the varying fraction of the contributing air streams. The authors might consider it worthwhile to add the air stream medians of those three variables as well to Fig. 11a-c, to see whether the air streams also change their properties over time or just the contributing fraction varies. Secondly, the causes for the change in the relative contribution of the different air streams remains unclear. Maybe the authors can either speculate upon the possible reasons for the temporal evolution, or declare this to be beyond the focus of the study?

- L340,341,342: I think you should correct all the "means" here to "medians", since your binned trajectory maps always display median properties if I understood it correctly. The same is true for later occurrences.
- L364: I think it would be the green line in Fig. 11d. But it might anyways be better to refer to Fig. 8 or 9a at this point, since Fig. 11 has not yet been discussed and appears considerably later in the manuscript.
- L364-366: The authors might want to reconsider the interpretation in this passage due to the following: First, are there really two air streams visible in Fig. 8 that merge over the Swiss Plateau? Especially considering the contour line in Fig. 9 it rather seems that one branch originates in the western part of the Swiss Plateau and another one in the central part of the Swiss Plateau, both of them merging approximately south of Zurich. Secondly, the figures don't explicitly depict a west foehn effect in the lee of the Swiss Jura. Since it is not shown explicitly, I think you should refrain from this statement, or declare it hypothetically.
- Fig. 10: At this point, the number of trajectories (7824 + 2672 = 10496) is again different to the number you mentioned in the methods Section of the manuscript. Could you comment (probably best in the methods) how you end up with this sample size (due to the exclusion of trajectories arriving higher than 1700 MSL?)?
- L440-442: This sentence could be rephrased in a more specific way to illustrate the linkage of upstream stability and the Alpine flow regime.

**Local aspects**

- L486: "Inn Valley boundary" → "IVB"
- Section 5.3: It's questionable whether this Section really provides additional insight for the main conclusions of the study, since similar processes have been discussed earlier on a regional scale in Section 4.2. Please motivate more clearly why the diabatic effects are discussed.

**Discussion**

- Corresponding to earlier comments and the synthesis: There is some discrepancy between the discussion and the results. While a considerable part of the results addresses the air mass origin and diabatic processes, the discussion solely focuses on the penetration mechanisms and the correct classification ("west foehn" vs. "north foehn"). Although there are probably no publications addressing diabatic processes of this foehn type, there are still a few publications addressing these aspects for other foehn events which might be considered useful as reference to embed the results into the literature in an additional, short discussion chapter.
- Section 6.2: The discussion of the foehn classification is rather detailed. Given that the Lagrangian perspective anyway indicates that the applicability of a binary classification is questionable, the authors might consider to shorten this aspect of the paper.
- L593: "Richter"  $\rightarrow$  "Richner"
- L610: "provide" → "provides"

**Conclusions**

• As a comment regarding the structure: The conclusion comprises an extensive list of eight bullet points. I think it would facilitate reader guidance if you could add some more structure to it, because it seems a bit unclear, if they are all equally important.

As a suggestion, they could be grouped according to the research question they address, and possibly the number could be reduced to one bullet point per research question/key result.

• After having read the conclusions, I still wonder whether the synoptic evolution of this case study represents a typical evolution for north or west foehn events. In other words, can similar results be expected for other northwest foehn cases? What changes do the authors expect for cases without orographic precipitation? Maybe you want to add one to two sentences addressing the generalization potential of the study.

---

## Author Response (AR1)

**Response to reviewers comments**
**Is it north or west foehn? A Lagrangian analysis of PIANO IOP 1**
**DOI: 10.5194/wcd-2021-65**

Manuel Saigger, Alexander Gohm

December 20, 2021

**1 Introduction**

We thank both reviewers for their detailed feedback and their suggestions to improve the manuscript. In the following we provide a point-by-point answer to all the reviewers' comments with the original comments in black and our responses in blue. In addition to the revised manuscript, we are providing a version in which all changes have been highlighted in blue (i.e. parts added) and in red (i.e. parts removed).

**2 Response to Reviewer 1**

**2.1 General Comments**

A case study of northwest or north foehn impacting the Inn Valley is presented. Foehn from this direction are much less common than southerly foehn for the Inn Valley and have not been examined in very many previous studies. Consequently, this study of such an event during the PIANO field campaign is timely and welcome. Overall, this case study is well presented and reasonably well written. There are a number of places where I think it is a little too detailed and parochial. I think during the revision I'd urge the authors to trim it down a little and focus on what will be of interest to community more broadly and more generally. I've indicated a few places where some information could be deleted or abbreviated without any impact on the key results or conclusions. I don't have any major comments but I have a number of specific, relatively minor, comments that I'd recommend the authors address.

We agree that the manuscript is rather long. We tried to make it more concise by removin parts especially in the methods and model validation (including Fig. 6).

**2.2 Specific Comments**

**Abstract**

Overall, the abstract seems quite long  does if conform to required word length?

We agree that the abstract is on the long side and slightly shortened it. We did not find any restrictions on the length of the abstracts in WCD. The current length (465 words) is comparable, e.g. to the WCD papers DOI: 10.5194/wcd-2-129-2021 (433 words) and DOI: 10.5194/wcd-1-45-2020 (432 words).

Line 1  reorder sentence , Austria, that occurred on 29 October 2017 is investigated...

We changed that.

Line 4  morning

We changed that.

Line 14  10-40 % of what?

We changed this sentence to "10 to 40 % of the trajectories..".

**Introduction**

Line 30  safety
We changed that.

L37  perhaps bracket (in the west-east aligned Inn Valley)
We think that the alignment of the valleys here and also in our study are quite important, so we would prefer to leave it that way.

L38 - rephrase Although it has been known... delete also
We changed that.

L45  replace about with of
We changed that.

L72  purely
We changed that.

L78 these methods
We changed that.

L83  you should probably cite Elvidge and Renfrew (2016, BAMS) who were the first to use trajectory analysis to quantify foehn warming mechanisms. Miltenberger et al. (2016) follow and cite that study.
We included that citation.

L84  time not times  were not was
We changed that.

L91-99  this paragraph is quite long  probably could be shortened.
This comment is in line with the suggestions of reviewer 2 to shorten this and the next paragraph. As the content of this paragraph gets not picked up later explicitly, we excluded the paragraph and shifted the references to Durran and Klemp (1983) and Zaengl and Hornsteiner (2007) to the end of the paragraph before.

**Section 2**

L127-132  this list of stations and heights could be deleted. The stations are already named in Figure caption 1 and the exact heights are not necessary. You could perhaps group them as mountain top or crest stations and valley stations in the caption.
We changed that.

L139-151  you dont need to tell us how you calculated potential temperature, or if you feel the height adjustment method is necessary just PT follows standard formula and a height adjustment is made as used by Muchinksi et al. But I dont think it is  that sort of adjustment is standard.
We shortened this paragraph and just mention which methods were used and when.

L191-196  the stuff about the CFL criteria could probably be deleted or abbreviated.
We shortened this paragraph and explain our choice of the integration time step when it is first mentioned two sentences before. However, our sensitivity experiments showed that the choice of the integration time step makes a difference and a small enough integration time step is necessary to fully resolve the vertical evolution. Therefore we want to keep a note on the time step.

L210-220  again the background on ice-liquid potential temperature could be abbreviated to one sentence and no formula.
As ice-liquid potential temperature is not that commonly used as a measure we think we should keep a large part of this paragraph explaining the meaning of $\theta_{il}$ and the interpretation of its changes. However, we shortened the last part on the approximations.

Note: you say 1000 m MSL etc many times, strictly speaking this should be 1000 m above MSL in every instance
Here we use MSL as the abbreviation for "above mean sea level" as suggested by the list of acronyms and abbreviations on the website of the American Meteorological Society: `https://www.ametsoc.org/index.cfm/ams/publications/author-information/formatting-and-manuscript-components/list-of-acronyms-and-abbreviations/`

**Section 3**

Figure 4 shows model cross-sections of a rotor in the Inn Valley at two times. I was wondering if there were any observations of rotors at this time? I know these were observed during PIANO (c.f. Haid et al. 2020). It would have been nice to have seen some of the lidar wind observations from PIANO used to validate the model in this way.
Unfortunately the lidar observations were located in the city center of Innsbruck, while the rotor is simulated in the Inn Valley west of Innsbruck, so we do not have observations to verify the simulated rotor. We could only use the lidar observations to verify the simulated wind structure above Innsbruck (see Fig. 5).

L303 I'd maybe rephrase, in Fig 7 the increase in wind speed and the sift to westerlies are both prominent, the speed increase perhaps more so.
That is true, we rephrased that sentence and included the increase in wind speed.

L324 I'd rephrase this line They should therefore not be seen as the truth in all aspects but rather as one possible flow realization to learn more about the air mass transport form a Lagrangian perspective. One flow realisation makes it sound like another simulation from say an EPS would capture something different, but I don't think that is the case. I think the limitations of the model simulation are more to do with model physical parameterizations (the cold-pool breakdown is common and related to BL turbulence etc). I'd suggest rephrasing to state the simulation is reasonably good and sufficiently good for detailed analysis of the origins of air masses and so on, i.e., the following analysis. But it has limitations most likely related to long- standing weaknesses most likely related to parameterization weaknesses.
We rephrased this section a bit.

**Section 4**

L331 Noting above, I'd rephase to reasonable agreement rather than good agreement
We corrected that.

L350-352 could perhaps be deleted doesn't add much
That is true, we deleted this.

L377 perhaps give the "times before arrival" of when these locations (Vosges etc) are passed
We included the times of the crossing of the Vosges and the Black Forest when Fig. 10a is first discussed in Section 4.1.

L394 I am afraid I could not see an increase in $\theta_{il}$ in Fig 10b in the median line. I can see the increases in theta that are discussed. Perhaps you mean in the shaded area? Similarly in Fig 10h, I struggled to see the gain in moisture the trace looked pretty flat to me. This does partly undermine some of this paragraph.
In Fig. 10b (now Fig. 9b) the increase in $\theta_{il}$ is not very strong for the Vosges and the Black Forest, but in my eyes still visible: starting at 6 h before arrival with about 298.5 K, increasing to slightly above 299 K around 3.5 h before arrival and to about 299.5 K at 2 h before the arrival. Concerning the gain in moisture, we were referring to the increase in the 10th percentile for the first three hours and than the increase in the median of $q_t$ from about 4.7 to 5.5 g kg$^{-1}$ between 3.5 to 2 h before the arrival.

Section 4.3 Temporal evolution In all honesty, I am not sure this section added much to the paper. It seems a bit of a detail to me. The foehn event is already quite short (a few hours) breaking it

into three phases seemed like more than readers would want to know to me. Id suggest discussing whether it is important. Or perhaps whether it would be better re-located to section 5.1 where the valley-crossing section shows some nice changes in time?

We agree that the last phase adds details that are not discussed later again. However, this phase is still part of the foehn event, so for completeness reasons it should be discussed here. Overall we would like to keep this section since also the second reviewer liked it. We tried to give a few more details on the connection between different air stream contribution and the overall changes of quantities and added a discussion paragraph where the changes on the upstream side are discussed. We think that the separation between the regional and local aspects as it is also done in the rest of the paper should remain, hence, a re-location to section 5.1 would not work. Nevertheless we tried to connect the two scales in a discussion paragraph where the possible connection between the upstream patterns and the penetration patterns is raised, however a definite answer on this connection is beyond the scope of this study.

**Section 5**

I thought Fig 12 was really good and illustrative of the changes in time. I found Fig 13 less interesting. The vectors are not really discussed, do they tell us anything other than the flow is along the valley? You mainly talk about the cross-valley flow (shading), while the theta contours are hard to see. I wonder if this figure could be improved or simplified or replaced?

We agree that some features in this figure shown by the vectors and the isentropes like the gravity wave structure and the rotor are already partly shown in Fig. 3 and 4. However, in our opinion here the connection between the Eulerian and Lagrangian perspective is done, which we would like to keep. Additionally the figure connects to the next figure on the potential impact of diabatic processes inside the valley.

Section 5.2 was also a bit long and started to get a little parochial. I wonder if youd be better trying to edit down these local findings and try to reframe this section into findings that are likely to be more generally applicable (e.g., to other locations or to other cases).

It is true that the description here is very concentrated on the local terrain features, however, we think that a detailed description is necessary to clearly identify the features we were talking about. With the section on the penetration mechanisms in the discussion we hope to give a broader view also for other cases and locations.

In general, I thought Sections 6 and 7 were very good, well done.

Thank you! The second reviewer had a couple of comments on these sections, so there are some changes in this as well (see comments below)

**3 Response to Reviewer 2**

**3.1 General Comments**

The authors present a case study of a west to northwest foehn event in the Inn Valley that occurred during the PIANO field campaign in Oct 2017. The mesoscale WRF simulation is validated using the extensive measurement dataset gathered during IOP1, and then combined with Lagrangian air parcel trajectories to assess three core topics:

- Air mass origin, air streams and diabatic processes upstream of the Alps

- Inn Valley boundary crossing and the associated penetration mechanisms

- The classification of such events into north or west foehn

The study is well-written and structured in a meaningful way. However, upon reading the title of the manuscript, I wonder whether it actually addresses the most important result of the study. In my opinion, Section 5.1 and 5.2 concerning the valley boundary crossing and the associated penetration mechanisms are an innovative assessment of Foehn descent mechanisms and pathways, combining the Lagrangian and the Eulerian perspective. The authors might consider this to be reflected in the title of the study as well. However, it is purely a suggestion and the importance of the key results can, of

course, be assessed differently. Besides this comment, I have some concerns regarding the verbosity of the manuscript. For example, while the large-scale overview and the WRF model validation is important to set the scene for the case study, it could be written in a more concise manner and, more directly, guide the reader towards the main conclusions of the study. Hence, I added some suggestions in the minor comments below, where I find the authors might consider re-writing and shortening certain text passages. In addition, I think the current embedment of diabatic processes into the study should be reconsidered. First of all, the respective research question could be rephrased more precisely. Secondly, the relevance of Section 4.2 and 5.3 remain, to a certain extent, unclear. Either, the authors emphasize the relevance of the discussion of diabatic processes for the penetration of the Foehn air, or embed the results into other literature addressing diabatic processes of Foehn flows. If so, I suggest adding an additional discussion chapter with respect to this topic. Or, alternatively, the results concerning diabatic processes could also be discarded from the study. I included the respective comments in the points below where I saw fit. Besides these minor concerns and some additional minor comments, I didnt identify any major concerns. Therefore, after the authors have addressed my comments, I would recommend the study for publication in *Weather and Climate Dynamics.*

We agree that our manuscript is quite long and detailed. As this was also a major criticism of reviewer 1 ,we tried to shorten and make the manuscript more concise. We especially shortened the sections on the methods and the model validation and removed Fig. 6.

We decided to keep the original title. The reviewer is right that we combine Lagrangian and Eulerian methods. However, the main results are based on the Lagrangian analysis. We believe that the current title better highlights this novel aspect compared to previous north foehn studies. The current title better fits with the content of the absract which mainly mentions the results of the Lagrangian analysis and stresses the limitation of the pure Eulerian approach.

We think that the role of the diabatic processes should be discussed in the study since it was an important process on the windward side and has been a point of conflict in the existing literature on this type of foehn in the Inn Valley. We added a new discussion part (section 6.1) in which we address flow patterns and diabatic processes (including orographic precipitation) on the upstream side. Indeed, it would be interesting to know how these diabatic processes determine the penetration of the flow into the valley. However, for answering this question a more comprehensive heat and momentum budget analysis would be required, which is beyond the scope of this study.

**3.2 Specific Comments**

**Abstract**

L10, L18: Terminology: When invoking the Lagrangian perspective, I would either always use the phrase air masses, air parcels, or simply air, but not use them as synonyms, in order to avoid terminological confusion. The authors might want to double check this throughout the manuscript and also clearly differentiate between air parcels and air stream (the latter being a coherent bundle of air parcels).

Thank you for pointing that out, we hope we found all the unclear instances.

L24-25: It is unclear to me what kind of foehn criteria you refer to here. Is north and west foehn distinguished based on an estimate of adiabatic heating? You could specify this with additional information, or re-write the last sentence of the abstract.

The method that we had in mind here would be similar to Plavcan et al (2014) where potential temperature difference to a crest station is one of the foehn criteria. However, we have removed this phrase as it is quite misleading and not mentioned elsewhere in the manuscript..

**Introduction**

Fig. 1: If I checked it correctly, the abbreviation IVB (Inn Valley boundary) is not defined where it is first used.

That is true, we changed that.

L77: "... various diabatic and adiabatic processes within the flow" sounds a bit strange to me. Maybe you could leave "within the flow" or replace it by "along the transport pathway".

We changed that.

L78: "this" → "these"
We changed that.

L79-L80: Could you specify a bit more what kind of events you refer to? "heat events" are probably "heat events in Japan", "melt events" are "melt events in polar regions"?
We provided more details.

L84-85: It seems a bit unclear to what location the 20 % refer to. To Innsbruck (i.e., the Inn Valley itself)? Or does this refer to upstream precipitation frequency? Please rephrase accordingly.
We made it more clear that in 20 % of the times there is precipitation in Innsbruck and in 40 % of the times there is precipitation upstream.

L91-105: As one suggestion to shorten the manuscript: Is this discussion of the impact of moisture on gravity waves really relevant, or could it be shortened?
The first reviewer also suggested to shorten this paragraph. As the content of this paragraph gets not picked up later explicitly, we removed the paragraph and shifted the references to Durran and Klemp (1983) and Zaengl and Hornsteiner (2007) to the end of the paragraph before. The content of the second part (L 99-105), however, is discussed later and therefore should in our opinion remain.

L115: Research question three could be formulated in a more precise manner. It remains unclear, whether the authors refer to the role of adiabatic and diabatic processes for the descent or the heat budget of air parcels. You might want to consider rephrasing the third goal of the study (see also related comment in the synthesis).
We emphasized that we are mainly interested in the budgets of heat and moisture along the trajectories.

**Data and methods**

Section 2.1 and Section 2.2 are both well-written, but rather extensive. Perhaps, the authors find one to two passages where they could shorten the methods section a bit, e.g., the listing of the measurements and stations used (L123-138), or the detailed description of the model setup (L160-169).
Reviewer 1 had a similar comment, hence, we shortened the paragraph on the stations sites, but we think that the rather detailed description of the model setup is necessary for reproducibility.

L188: I dont understand how the total number of trajectories can be only 18688. If you have dx=dy=200 m on a 4x4 km square, this would result in 21 x 21 = 441 trajectories per horizontal plane. Combining it with dz = 25 m over a distance of 2500m 700m = 1800m, I end up with 441*73 = 32193 trajectories. Maybe I misunderstood the trajectory setup. In that case, I would be glad if you can clarify it shortly.
Thank you for checking so much in detail and finding errors we missed. Due to a mistake in the setup of the starting positions we actually only have a 3 km wide square which results in 265 trajectories per layer and 18688 trajectories in total.

L191-192: Could you please explain what this jump option implies?
We added a short sentence to explain the jump option (trajectories intersecting with the topography are lifted above the topography and are allowed to continue. If this is not done, about half of the trajectories are lost.)

L200: Please give some references to these other trajectory tools.
We added references to FLEXPART and HYSPLIT.

L202-203: I think I understand what you mean but the sentence seems a bit hard to understand at first. Maybe rephrase in an easier way.
We tried to rephrase to make it clearer what we mean.

L210: Are there not five hydrometeor classes in the microphysics scheme (cloud water, rain, cloud ice, snow, graupel)?
That is correct, we changed this.

L235-L240: I had difficulties in understanding this entire passage. I think you should consider reformulating the description of the Eulerian mapping of Lagrangian trajectory information. Also, Im unsure whether you calculate mean trajectory properties or median properties, since you write by taking the median of various parameters.
We tried to clarify the phrasing in this paragraph.

**Meteorological overview**

Section 3 is, as well, rather extensive considering it is not addressing any research question directly. However, the meteorological overview is still helpful to understand the case study and, therefore, it is fine to have an extensive overview. However, I added one suggestion to shorten a passage below.
We agree that the overview of the event and the model validation are quite important for the understanding of the event.

Fig. 5: With the chosen colorbar I have some difficulties inferring on the vertical stratification from the figure. Maybe a few contour lines if potential temperature could be added for readability?
We added isentropes each 2 K.

L269: Since the flow deflection is not visible very clearly at 700 hPa, you could add "(not shown)" to the first part of the sentence. Or, alternatively, display the 850 hPa wind field in Fig. 2b.
We corrected that.

Fig. 7 caption: The phrasing "... with values every integration time step at the WRF grid point closest to UNI" confuses me a bit. Probably you rather mean "... with values from every output time step of the WRF grid point closest to UNI."?
In that context the phrasing is actually correct, as we use the time series output of WRF (https://github.com/wrf-model/WRF/blob/master/run/README.tslist) so we have the current values at each integration time step. However, we tried to clarify the phrasing.

L298-315: Could this be more concise since the comparison just serves for the model validation?
We agree that the aspect of the along-valley foehn breakthrough and breakdown is not really in the focus of the study, so we removed it. As a consequence Fig. 6 was also removed.

L296: typo ("form" → "from"). I saw this typo multiple times, please check at other places throughout the manuscript.
We corrected that.

L318: cold-air pool → CAP
We corrected that.

L324-325: The meaning of this sentence is a bit hard to grasp. Maybe you could rephrase it or explain it a bit more precisely.
This sentence was also criticized by the other reviewer and we rephrased this and the sentence before.

**Air mass origin and history**

Section 4.2 As mentioned in the synthesis, the goal of this Section remains a bit unclear What do we learn from it? Is it relevant for the penetration into the Inn Valley? In my opinion, it would need a bit more context why the thermodynamic assessment is of relevance for the foehn community (e.g., in an additional, short discussion chapter).
We tried give a bit more context to the regional scale analysis with an additional discussion chapter. We agree that the connection between the regional scale and the valley scale would be very interesting, though in the context of our study quite speculative. We tried to point this out with one additional

paragraph in the discussion of the penetration mechanisms.

Section 4.3: The Section is very interesting, since it aims to explain the differing thermodynamic properties and altitude of origins by the varying importance of two different air streams. However, the reader is left with two open questions at this point: First of all, since the total moisture change, the altitudes and the total potential temperature change is shown for all trajectories, their temporal evolution can only indirectly be linked to the varying fraction of the contributing air streams. The authors might consider it worthwhile to add the air stream medians of those three variables as well to Fig. 11a-c, to see whether the air streams also change their properties over time or just the contributing fraction varies. Secondly, the causes for the change in the relative contribution of the different air streams remains unclear. Maybe the authors can either speculate upon the possible reasons for the temporal evolution, or declare this to be beyond the focus of the study?
We tried out your suggestion, however especially subpanel (a) gets very messy in that case. Nevertheless, thank you for pointing out the missing connection. Hence, we added a short sentence on the contribution of the southern air stream. As you point out, the cause for the change in the air stream contributions remains speculative, therefore we think that is better located in the discussion chaper.

L340,341,342: I think you should correct all the "means" here to "medians", since your binned trajectory maps always display median properties if I understood it correctly. The same is true for later occurrences.
That is true, we changed this.

L364: I think it would be the green line in Fig. 11d. But it might anyways be better to refer to Fig. 8 or 9a at this point, since Fig. 11 has not yet been discussed and appears considerably later in the manuscript.
We agree with your comment, however, we think that here it also would be better to refer to Fig 9 (former Fig.10).

L364-366: The authors might want to reconsider the interpretation in this passage due to the following: First, are there really two air streams visible in Fig. 8 that merge over the Swiss Plateau? Especially considering the contour line in Fig. 9 it rather seems that one branch originates in the western part of the Swiss Plateau and another one in the central part of the Swiss Plateau, both of them merging approximately south of Zurich. Secondly, the figures dont explicitly depict a west foehn effect in the lee of the Swiss Jura. Since it is not shown explicitly, I think you should refrain from this statement, or declare it hypothetically.
The foehn in the lee of the Swiss Jura can be seen in Fig. 9a and h, however, as this is never discussed again and not relevant for the outcome of the study, we removed this sentence.

Fig. 10: At this point, the number of trajectories (7824 + 2672 = 10496) is again different to the number you mentioned in the methods Section of the manuscript. Could you comment (probably best in the methods) how you end up with this sample size (due to the exclusion of trajectories arriving higher than 1700 MSL?)?
See comment above for Sect. 2.3.1.

L440-442: This sentence could be rephrased in a more specific way to illustrate the linkage of upstream stability and the Alpine flow regime.
We tried to make the connection clearer.

**Local aspects**

L486: "Inn Valley boundary" → "IVB".
We changed this.

Section 5.3: It's questionable whether this Section really provides additional insight for the main conclusions of the study, since similar processes have been discussed earlier on a regional scale in Section 4.2. Please motivate more clearly why the diabatic effects are discussed.
We think that the structure regional → local should be kept and a second focus on the diabatic processes on the leeward side is important, as this is one of the main conflicts in the existing literature

on north foehn. However, we tried to make the motivation a bit more clear.

**Discussion**

Corresponding to earlier comments and the synthesis: There is some discrepancy between the discussion and the results. While a considerable part of the results addresses the air mass origin and diabatic processes, the discussion solely focuses on the penetration mechanisms and the correct classification ("west foehn" vs. "north foehn"). Although there are probably no publications addressing diabatic processes of this foehn type, there are still a few publications addressing these aspects for other foehn events which might be considered useful as reference to embed the results into the literature in an additional, short discussion chapter.

Thank you for pointing out this lacking discussion, as mentioned in the comments earlier, we added a section where we discuss the upstream processes.

Section 6.2: The discussion of the foehn classification is rather detailed. Given that the Lagrangian perspective anyway indicates that the applicability of a binary classification is questionable, the authors might consider to shorten this aspect of the paper.

It is true that in the end our conclusion is that a binary classification is questionable. However, in the existing literature on this kind of event the different classification approaches show the gap in knowledge on the event. Therefore, we want to propose our new approach while still pointing out that the whole idea of a strict classification may not be purposeful.

L593: "Richter" → "Richner"
We changed that.

L610: "provide" → "provides"
We changed that.

**Conclusions**

As a comment regarding the structure: The conclusion comprises an extensive list of eight bullet points. I think it would facilitate reader guidance if you could add some more structure to it, because it seems a bit unclear, if they are all equally important. As a suggestion, they could be grouped according to the research question they address, and possibly the number could be reduced to one bullet point per research question/key result.

In our case the bullet points of the conclusion reflect the structure of the paper of general → regional scale → local scale. We tried to clarify this structure by breaking up the bullet points into this structure. Structuring it according to the research questions would not really work here, as some research questions are answered for several different scales.

After having read the conclusions, I still wonder whether the synoptic evolution of this case study represents a typical evolution for north or west foehn events. In other words, can similar results be expected for other northwest foehn cases? What changes do the authors expect for cases without orographic precipitation? Maybe you want to add one to two sentences addressing the generalization potential of the study.

We added a few lines about the generalization potential here and at the end of the discussion of the foehn penetration. Although the observed patterns are in agreement to pattern that have been observed earlier at different locations, we think a generalization is impossible as it was shown here that quite subtle changes in the crest-level conditions are connected to strong changes in the penetration patterns.

---

## Author Response (AR2)

**Response to reviewers comments**

**Is it north or west foehn? A Lagrangian analysis of PIANO IOP 1**
DOI: 10.5194/wcd-2021-65

**Manuel Saigger, Alexander Gohm**

**January 26, 2022**

Thank you for accepting the manuscript for publication. Following the recommendation we went through the manuscript and shortened a few passages, where we think that this is possible without losing context. However, this did not change the overall character of the manuscript. We additionally provide a version of the manuscript where these changes are highlighted. Our changes are:

- Section 4.1, Lines 311–320: deleted rough description of the vertical evolution as this is described in more detail for the individual trajectory branches in the next two paragraphs in more detail. Added information in the last paragraph (Lines 330–335) which would have been lost otherwise.

- Additionally updated Figures 7 and 9 to simplify the information on the air stream contributions.

- Rearranged Section 4.2.1 to avoid repetitions inside the paragraph.